# Dorsal striatum coding for the timely execution of action sequences

**Maria Cecilia Martinez[1,2], Camila Lidia Zold[2,3], Marcos Antonio Coletti[2,3], Mario Gustavo Murer[2,3]\*[†], Mariano Andrés Belluscio[2,3]\*[†]**

[1]Universidad de Buenos Aires, Facultad de Ciencias Exactas y Naturales, Departamento de Fisiología, Biología Molecular y Celular "Dr. Héctor Maldonado", Buenos Aires, Argentina; [2]Universidad de Buenos Aires - CONICET, Instituto de Fisiología y Biofísica "Dr. Bernardo Houssay" (IFIBIO-Houssay), Grupo de Neurociencia de Sistemas, Buenos Aires, Argentina; [3]Universidad de Buenos Aires, Facultad de Ciencias Médicas, Departamento de Fisiología, Buenos Aires, Argentina

**Abstract** The automatic initiation of actions can be highly functional. But occasionally these actions cannot be withheld and are released at inappropriate times, impulsively. Striatal activity has been shown to participate in the timing of action sequence initiation and it has been linked to impulsivity. Using a self-initiated task, we trained adult male rats to withhold a rewarded action sequence until a waiting time interval has elapsed. By analyzing neuronal activity we show that the striatal response preceding the initiation of the learned sequence is strongly modulated by the time subjects wait before eliciting the sequence. Interestingly, the modulation is steeper in adolescent rats, which show a strong prevalence of impulsive responses compared to adults. We hypothesize this anticipatory striatal activity reflects the animals' subjective reward expectation, based on the elapsed waiting time, while the steeper waiting modulation in adolescence reflects age-related differences in temporal discounting, internal urgency states, or explore–exploit balance.

**\*For correspondence:**
gmurer@fmed.uba.ar (MGM);
mbellu@fmed.uba.ar (MAB)

[†]These authors contributed equally to this work

## Editor's evaluation

This article investigates an important topic related to the initiation signals of actions sequences detected in the dorsal striatum. The data presented convincingly support the idea that these signals distinguish between the premature versus the timely release of actions. The experiments are well-organized and substantially advance the field.

## Introduction

The striatum is involved in the acquisition and execution of action sequences (*Costa, 2011*; *Graybiel, 1998*; *Hikosaka et al., 1999*). It has also been linked to temporal information processing (*Bakhurin et al., 2017*; *Emmons et al., 2017*; *Gouvêa et al., 2015*; *Matell et al., 2003*; *Mello et al., 2015*) and to the use of temporal information for the action initiation timing (*Kunimatsu et al., 2018*; *Thura and Cisek, 2017*; *Yau et al., 2020*). Sometimes, well-learned action sequences need to be initiated at precise times to obtain the desired outcome. However, these actions may be difficult to withhold when triggering cues are present, and also difficult to stop once they have been initiated (*Dalley and Robbins, 2017*; *Gillan et al., 2016a*; *Graybiel, 2008*; *Knowlton and Patterson, 2016*; *Robbins and Costa, 2017*). Moreover, in neuropsychiatric conditions involving malfunctioning of cortico-basal ganglia circuits, such as attention deficit hyperactivity disorder, Tourette syndrome, obsessive–compulsive disorder, and drug addiction (*Dalley and Robbins, 2017*; *Gillan et al., 2016a*; *Singer, 2016*)

action sequences might be started at inappropriate contexts and timings. However, how the striatum contributes to action sequence initiation timing remains poorly understood.

Interestingly, impulsivity has been identified as a vulnerability factor for compulsive drug seeking habits (*Belin and Everitt, 2008*). Recent studies link impulsivity to automaticity in behavior (*Ersche et al., 2019*; *Gillan et al., 2016b*; *Hogarth et al., 2012*) and a preponderance of habitual over goal-directed behavioral control (*Everitt et al., 2008*; *Voon et al., 2015*). An influential theory in the field postulates a dual control system for behavior, where dorsolateral striatal (DLS) circuits support habitual stimulus–response control whereas dorsomedial striatal (DMS) circuits mediate cognitive-based deliberative control (*Balleine and Dickinson, 1998*; *Daw et al., 2005*; *Graybiel, 2008*; *Yin and Knowlton, 2006*). During action sequence learning, neuronal activity in the DLS rapidly evolves to mark the initiation and termination of the acquired sequence (*Jin and Costa, 2010*; *Jog et al., 1999*), possibly contributing to its release as a behavioral unit or chunk (*Graybiel, 2008*). In contrast, the DMS encodes reward expectancy, reward prediction errors, and trial outcomes even after extensive training (*Kim et al., 2009*; *Kubota et al., 2009*; *Rueda-Orozco and Robbe, 2015*; *Samejima et al., 2005*; *Thorn et al., 2010*; *Vandaele et al., 2019*). All these features likely contribute to the regulation of explore–exploit (*Barnes et al., 2005*), cost–benefit (*Floresco et al., 2008*; *Schultz, 2015*), and speed–accuracy (*Thura and Cisek, 2017*) tradeoffs during decision making. A bias toward exploration and risk-taking (*Addicott et al., 2017*), low tolerance for delayed rewards (*Dalley and Robbins, 2017*; *Monterosso and Ainslie, 1999*; *Wittmann and Paulus, 2008*) and elevated internal urgency states (*Carland et al., 2019*), may all contribute to impulsivity traits. However, how neuronal activity in the dorsal striatum informs about enhanced premature responding in conditions of high impulsivity, remains unknown.

Here, we studied striatal activity during a self-paced task where rats have to withhold a rewarded action sequence until a waiting interval has elapsed (*Zold and Hussain Shuler, 2015*). Prematurely initiated sequences were penalized by re-initiating the waiting interval. Despite the negative effect this timer re-initialization may have, the animals showed premature responding even after extensive training. Moreover, they often failed to interrupt the execution of the action sequence although there was sensory evidence of its untimely initiation. Thus, the task allowed comparing striatal activity during behaviorally indistinguishable prematurely and timely executed learned action sequences. We found a peak of striatal activity preceding trial initiation that was modulated by the time waited before responding. In addition, this modulation grew at a faster rate in adolescent rats. This likely reflects a steeper growing increase in reward expectancy during waiting that could underlie their more impulsive behavior compared to adult rats.

## Results
### Rats learn to make timely action sequences to obtain a water reward

Water-deprived rats were trained to obtain water from a lick tube located within a nose-poke by emitting a sequence of eight licks following a visual cue (*Figure 1A, B*; task modified from *Zold and Hussain Shuler, 2015*). Animals self-initiated the trials by entering into the nose-poke. In those trials initiated 2.5 s after the end of the previous trial (timely trials), a 100-ms duration visual cue (two symmetrical green LEDs located at the nose-poke sides) reported that there was a 0.5 probability of receiving a water reward. In contrast, prematurely initiated trials (<2.5 s waiting time) were penalized by re-initiating the timer and had no visual cue associated with them. Spike discharges and local field potentials (LFPs) were recorded from the dorsal striatum using custom-made tetrodes (*Vandecasteele et al., 2012*) (representative localization *Figure 1C*, for detailed localization see *Figure 1—figure supplement 1*).

Adult rats learned to make timely nose-poke entries followed by an 8-lick sequence (*Figure 1D*). This was evidenced by a twofold increase in the reward rate in sessions late in training (after three consecutive sessions with >70% correct trials) compared to early in training (*Figure 1E*, P=6.10 × 10$^{-5}$, Wilcoxon matched pairs test, $n$ pairs = 15). Performance in 8-lick trials became faster with training (*Figure 1F–H*): trial duration (learning stage: $F_{(1, 14)}$ = 19.85, p = 5.43 × 10$^{-4}$, trial: $F_{(1, 14)}$ = 19.91, p = 5.37 × 10$^{-4}$), latency to the first lick (learning stage: $F_{(1, 14)}$ = 12.59, p = 3.21 × 10$^{-4}$), and time to complete the 8-lick sequence (learning stage: $F_{(1, 14)}$ = 18.33, p = 7.60 × 10$^{-4}$), diminished with training for both rewarded and unrewarded timely trials (significant effect of learning stage, non-significant interaction,

**Figure 1.** Rats become skilled in the task. (**A**) Behavioral chamber with the nose-poke. Animals' entries/exits from the nose-poke and licks are detected with infra-red (IR) beams. A 100-ms visual cue is presented through a pair of green LEDs placed on the side of the nose-poke to indicate a timely entry. (**B**) Schematic representation of the different types of trials. Timely trials require a minimum waiting time of 2.5 s and premature trials are those in which the minimum waiting time is not met. After that, trials are classified by whether animals performed an 8-lick sequence or not. (**C**) Top: Representative diagram of the electrodes' positioning, aimed at the dorsal striatum. Bottom: histological section (AP = 0.24 cm from bregma) with electrodes traces. (**D**) Percentage of the different trial types per session, timely trials with an 8-lick sequence (T×8L) or not (T<8L) and premature trials with an 8-lick sequence (P×8L) or not (P<8L). (**E**) Reward rates for early and late training stages. Trial duration (**F**), latency to the first lick during correct trials (**G**), and time to complete the 8-lick sequence (**H**) for the two types of correct trials (rewarded and unrewarded), at each training stage. (E–H) Data are expressed as mean ± standard error of the mean (SEM), n = 15 early and 15 late sessions, n = 5 rats performing the standard 2.5 s waiting time task,*** p<0.001; ** p<0.01. (**I**) Raster plot of the licks from 50 trials at a late training session of one of the adult rats. Colored dashes show the first 8 licks of the trial (timely rewarded trials: blue, timely unrewarded trials: lilac, premature trials: red).

The online version of this article includes the following figure supplement(s) for figure 1:

**Figure supplement 1.** Localization of recording tetrodes.

two-way Repeated Measures ANOVAs). A representative raster plot showing licking bouts during a whole session is shown in *Figure 1A, I*, and a representative video recording of behavior displayed during premature and timely trials is shown in *Video 1*.

Premature nose-poke entries delayed the opportunity to get the reward as evidenced by a negative correlation between the relative number of premature trials and reward rate across sessions (*Figure 2A*, $R^2 = 0.090$, $F_{(1, 60)} = 5.96$, p = 0.018, n = 62 sessions from 5 rats). Premature trials diminished with training from >30% of all trials at the beginning of training to ~15% at the

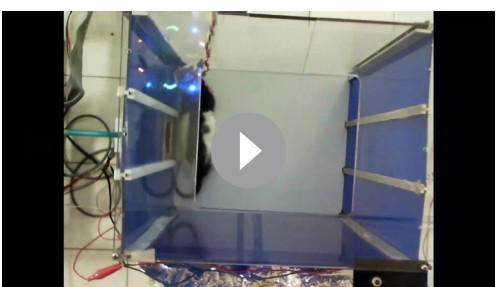

**Video 1.** Behavior in a late training session of an adolescent rat.

https://elifesciences.org/articles/74929/figures#video1

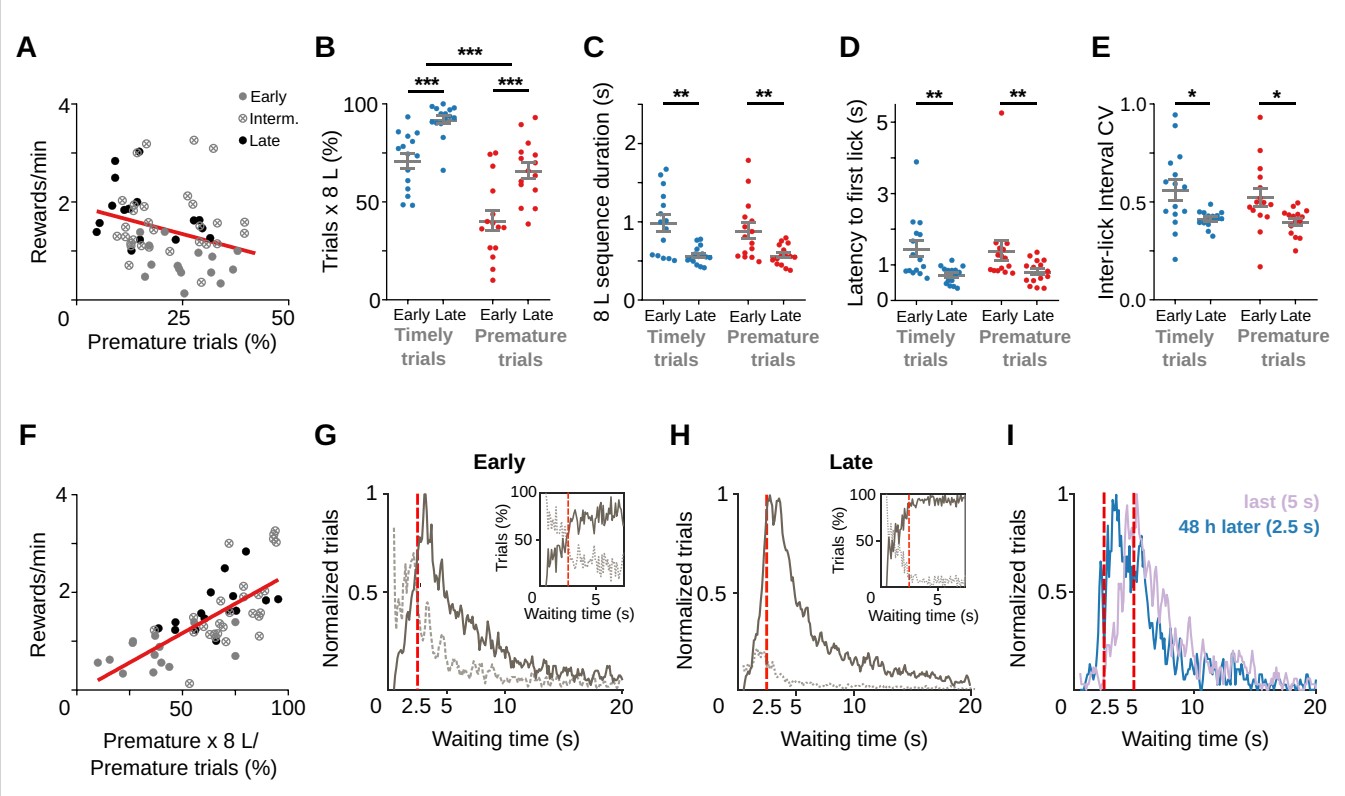

**Figure 2.** Training does not suppress premature initiations of the learned behavioral response. Data correspond to five adult animals performing the standard 2.5 s waiting time task. (**A**) Correlation between all prematurely initiated trials and reward rate ($Y = -0.023*X + 1.921$, slope significantly different from zero p = 0.018, n=62 sessions from a total of 5 rats). The proportion of trials (**B**), time to complete the 8-lick sequence (**C**), latency to first lick (**D**), and coefficient of variation (**E**) of the 8-lick sequence inter-lick intervals for prematurely and timely initiated trials, at early and late learning stages. (**B–E**) Data are expressed as mean ± standard error of the mean (SEM), n = 15 early and 15 late sessions, *** p<0.001 ; ** p<0.01; * p<0.05. (**F**) Correlation between percentage of prematurely initiated trials followed by an 8-lick sequence and reward rate ($Y = 0.024*X - 0.0408$, slope significantly different from zero, p = $1.03 \times 10^{-9}$, n=62 sessions from a total of 5 rats). (**G–I**) Normalized frequency distributions of the trial initiation times (waiting time), separated for trials with (gray solid line) and without (dotted line) 8-lick sequences and for early (**G**) and late (**H**) training stages (n = 15 sessions). Insets: percentage of the trials with (gray solid line) and without (dotted line) 8-lick sequences for each bin, zoomed around the criterion time. (**I**) Rats were trained with a 5-s criterion time period (blue) and afterwards were switched to a 2.5-s criterion time for the following two sessions (48 hr after the last 5-s waiting time session, lilac). Red dashed lines: criterion time; bin size: 100 ms; reference for normalization: bin with highest value = 1.

The online version of this article includes the following figure supplement(s) for figure 2:

**Figure supplement 1.** Rats trained with a longer waiting time also emit premature learned responses.

**Figure supplement 2.** Rats also learn a task with lower and upper limits in the waiting time.

end of training. However, premature trials followed by an 8-lick sequence rose from 40% to 70% of all premature trials with training, paralleling the relative increase of 8-lick sequences observed in timely trials (*Figure 2B*, $F_{(1, 14)} = 32.87$, p = $5.20 \times 10^{-5}$ for learning stage and $F_{(1, 14)} = 121.74$, p = $2.73 \times 10^{-8}$ for trial timing, no interaction, two-way RM ANOVA). These data suggest that behavior during premature trials was modified by learning, just like in timely trials. Further supporting this presumption, time to complete the 8-lick sequence ($F_{(1, 14)} = 15.63$, p = 0.001), latency to the first lick of the sequence ($F_{(1, 14)} = 14.30$, p = 0.002) and variation coefficient of the inter-lick intervals (an index of the regularity of such intervals; $F_{(1, 14)} = 6.78$, p = 0.021), diminished with training both for timely and premature trials (*Figure 2C–E*; significant effect of learning stage, no effect of trial type, no interaction, two-way RM ANOVA). Remarkably, even though premature trials were detrimental to reward rate (*Figure 2A*), there was a significant positive correlation between the percentage of premature trials followed by an 8-lick sequence and reward rate (*Figure 2F*, $R^2 = 0.478$, $F_{(1, 60)} = 54.98$, p = $1.03 \times 10^{-9}$, n = 62 sessions from 5 rats).

To characterize the timing of trial initiations in adult rats, plots showing the frequency distribution of all trial initiation times were built (*Figure 2G, H*). The probability of a trial including an 8-lick sequence

sharply increased at the end of the 2.5 s waiting interval, peaked immediately after its finalization, and then diminished gradually. Similar results were observed in a separate group of rats trained with a longer waiting interval (*Figure 2—figure supplement 1*). Finally, rats trained with the long waiting interval (5 s) quickly learned to adjust trial initiations to a shorter waiting interval (2.5 s) (*Figure 2I*), suggesting that premature trials with 8-lick sequences served to adapt behavior to changes in the waiting time requirements of the task that otherwise would have passed unnoticed to the rats.

An additional group of adult rats was trained with a modified version of the task that required initiating trials not before 2.5 s and no later than 5 s after exiting the port in the previous trial. These animals showed similar behavior with a faster decrease of trial initiations after the 2.5 s waiting time (see below; *Figure 2—figure supplement 2*).

Altogether, the data show that adult rats improved their reward rate by waiting the least possible time between trials. Noteworthy, premature execution of the learned behavioral response was (relatively) more common late in training than early in training, suggesting that with training behavior became less sensitive to the absence of the reward-predictive visual cue.

## Task-sensitive striatal activity concentrates at the boundaries of the learned behavioral response

To determine if striatal activity marks the boundaries of the learned action sequence in our task (*Jin and Costa, 2010*; *Jog et al., 1999*), dorsal striatum activity recorded from adult rats was analyzed by aligning the activity to port entry and port exit (*Figure 3A–C*). Visual inspection of neuronal raster plots and peri-event time histograms (PETHs) showed strong modulations of striatal activity preceding port entry and/or at the time of port exit, during timely trials (*Figure 3A, B*). Overall, ~50% of the recorded units (473 out of 867) showed higher activity (>1 SD over baseline) preceding port entry ('anticipatory activity'; 19%, *Figure 3D*), at the time of port exit (18%, *Figure 3E*) or both before port entry and at port exit (12%, *Figure 3F*). On average, these neurons showed lower than baseline firing rates during the execution of the learned action sequence (*Figure 3D–F*). There were also neurons ($n$ = 108, 12% of all recorded neurons) showing higher activity when the animal was inside the port than during the waiting period or at the initiation and finalization of the action sequence (*Figure 3G*). Finally, 82 neurons classified as non-task responsive were tonically active during the waiting period regardless of the waited time (*Figure 3—figure supplement 1A*). All main types of task-related activity emerged early during training (*Figure 3—figure supplement 1B*).

Thus, although striatal activity was continuously modulated during the present task, modulations at the boundaries of the behavioral response accounted for about 50% of all task-related activity and more than 30% of the recorded neurons showed a peak of activity anticipating trial initiation.

## Waiting time modulates anticipatory activity

Striatal activity that anticipates a learned behavioral response could specifically mark the initiation of a previously rewarded action sequence (*Jin and Costa, 2010*; *Jog et al., 1999*; *Martiros et al., 2018*). Likewise, it could relate to additional factors, such as reward anticipation or the vigor and value of an upcoming action (*Lauwereyns et al., 2002*; *Samejima et al., 2005*; *Wang et al., 2013*). Moreover, it has been proposed that changes in striatal activity preceding the initiation of a prepotent action may predict premature responding (*Buckholtz et al., 2010*; *Donnelly et al., 2014*; *Wu et al., 2018*). Because in the present task the learned action sequence is often prematurely released, we asked if the observed anticipatory activity could specifically predict its release, and if, additionally, encodes its timing. When all port entry responsive neurons were considered (i.e., port entry only plus port entry/port exit neurons), the average firing rate modulation anticipating trial initiation was higher for timely than for premature trials irrespective of the upcoming action including the 8-lick sequence or not (*Figure 4A, B*, $F_{(1, 330)}$ = 30.84, p = 7.01 × $10^{-8}$, significant main effect of trial initiation timing, no effect of action sequence-structure $F_{(1, 330)}$ = 0.089, p = 0.765, no interaction, $F_{(1, 330)}$ = 9.64 × $10^{-5}$, p = 0.992, two-way RM ANOVA). On average, this activity began 1 s before and peaked 0.5 s before the animal crossed the infrared beam located at port entry, both in premature and timely trials (*Figure 4A*). Since this anticipatory activity closely preceded approaching movements toward the nose-poke, we analyzed accelerometer recordings of head movements available from two adult rats. The accelerometer recordings did not show differences in movement initiation time between premature and timely 8-lick trials (0.357 ± 0.027 and 0.382 ± 0.023 s before port entry– mean and standard error of

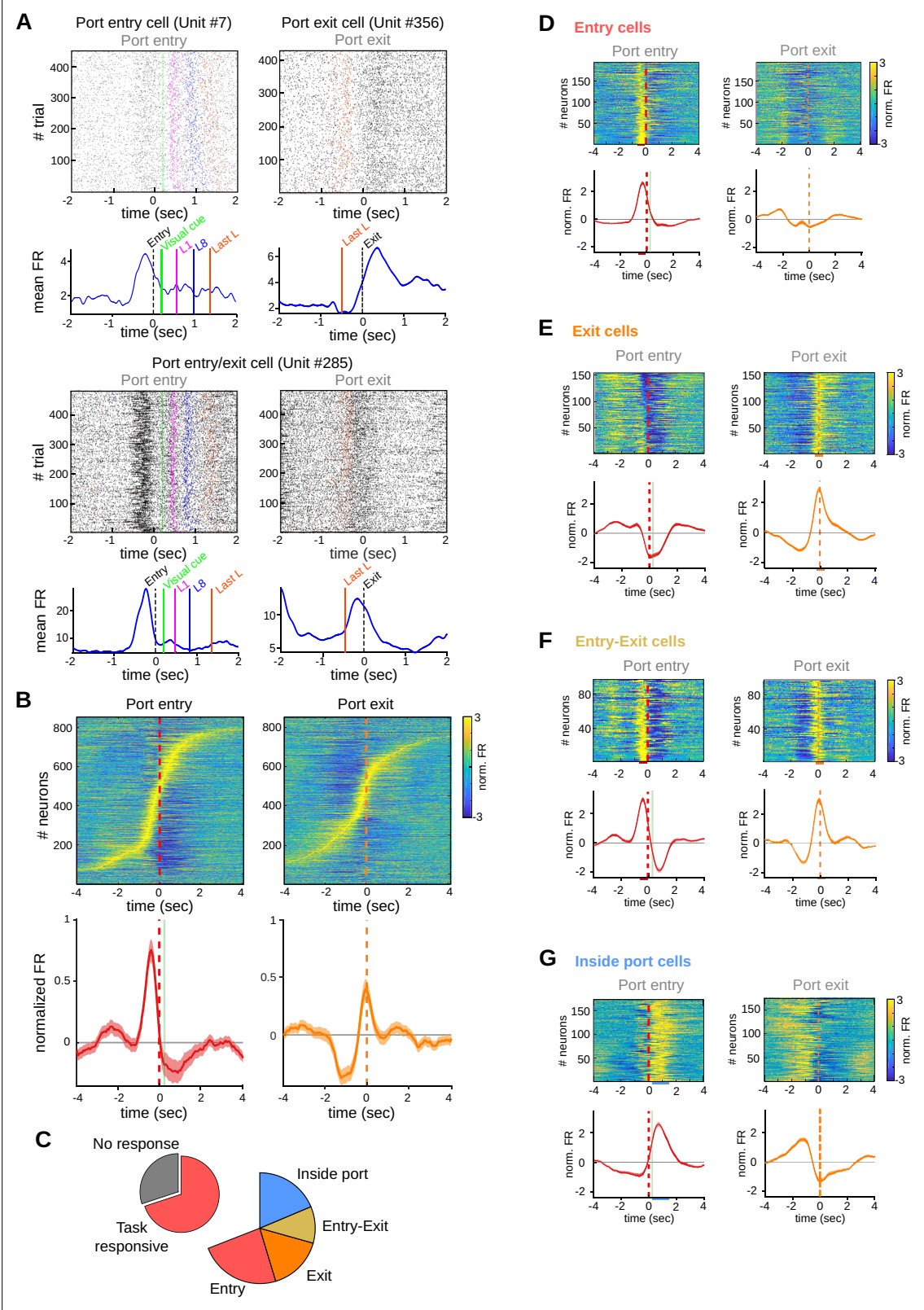

**Figure 3.** Striatal activity marks transitions between behavioral states of the task. Data come from five adult animals performing the standard 2.5-s waiting time task and two adult animals performing the 2.5–5 s cutoff version of the task. (**A**) Representative raster plots (above) and peri-event time histogram (PETH) (below) of striatal units showing firing rate modulations related to port entry and/or port exit. L1: first lick, L8: eighth lick, Last L: last lick. (**B**) Individual PETHs of striatal neurons during correct trials, aligned to port entry (left) or port exit (right). Below: average PETH (solid line) and

*Figure 3 continued on next page*

*Figure 3 continued*

standard error of the mean (SEM) (shaded area) for all recorded neurons. Red dashed line: port entry, orange dashed line: port exit, vertical green line: led on. (**C**) Proportion of neurons showing task-related firing rate modulations. PETH for all individual neurons and population average PETH aligned to port entry (left panels) and to port exit (right panels) for (**D**) striatal neurons showing only port entry-related activity, (**E**) striatal neurons showing only port exit-related activity, (**F**) striatal neurons showing activity modulations at both port entry and port exit, and (**G**) striatal neurons showing higher activity while animals are inside the port. From (**D–G**), data are the mean (solid lines) and SEM (shaded area). Colored bars over the *x*-axis show the interval used to detect firing rate modulations (red: entry, orange: exit, light-blue: inside port).

The online version of this article includes the following figure supplement(s) for figure 3:

**Figure supplement 1.** Characteristics of the neurons registered in the 2.5-s waiting time task.

the mean (SEM) – for premature and timely trials, respectively, $t_{(8)}$ = 0.636, p = 0.543, Paired *t*-test; *Figure 4C*). Furthermore, the 8-lick sequences emitted during premature and timely trials lasted the same and had the same latency and inter-lick interval regularity (*Figure 2C–E*), suggesting similar action vigor during premature and timely 8-lick trials. Thus, although head acceleration data may not include differences in movements of other body parts, our data support that in this task, the firing rate modulation preceding trial initiation discriminates between premature and timely trials and does not predict the speed, regularity, structure, value, or vigor of the subsequently released action sequence.

To further investigate this anticipatory activity, we plotted its amplitude at increasing waiting times observing that it increased with a steep slope as time waited reached the learned waiting interval and then plateaued (*Figure 4D, E*). Similar results were obtained in rats trained with a longer waiting interval. Because of the longer waiting time, behaviour becomes less organized during the first seconds after port exit in the 5s task, however, the modulation of activity is still observed in the bins that are close to port entry (*Figure 4—figure supplement 1A*). In contrast, there was no modulation of this neuronal activity by time waited in the following trial (*Figure 4F*), nor of port exit-related activity by the previous waiting time (*Figure 4G*). The steep slope of the curve at the criterion waiting time suggested that the neuronal activity does not linearly report elapsed time but is rather related to changes in reward anticipation as waiting progressed. To explore this possibility, we analyzed striatal activity of rats trained with a modified version of the task requiring initiating trials not before 2.5 s and no later than 5 s after exiting the port in the previous trial (*Figure 2—figure supplement 2*). Waiting less than 2.5 s (premature trials) or more than 5 s (late trials) was penalized by resetting the waiting timer (*Figure 4H*). We reasoned that if this activity provides a reward anticipation signal for the upcoming action, it should decrease after 5 s of waiting in the modified version of the task, considering that trials with >5-s waiting time are not rewarded. As in the standard version of the task, the animals learned to wait the less possible time between trials, and also noticed the effect of the cutoff time on reward probability, as evidenced by a reduced number of late trials with training (*Figure 4I*, $q_{(20)}$ = 10.64, p = 1.10 × 10$^{-9}$ versus timely trials, Tukey post hoc test after significant one-way RM ANOVA [$F_{(1.70, 34.05)}$ = 33.07, p = 1.81 × 10$^{-6}$]; *Figure 2—figure supplement 2*). The firing rate modulation preceding trial initiation increased with a steep slope at 2.5 s, but instead of plateauing, it decreased after surpassing the 5-s cutoff time (*Figure 4J*). A comparison of the effects of waiting time on anticipatory activity in the tasks with and without the 5-s cutoff yielded a significant interaction in a two-way RM ANOVA (*Figure 4K*, significant effect of task, $F_{(1, 334)}$ = 4.54, p = 0.034; trials, $F_{(1.96, 656.34)}$ = 67.73, p = 1.01 × 10$^{-15}$; interaction $F_{(2, 668)}$ = 29.27, p = 6.53 × 10$^{-13}$, two-way RM ANOVA).

In summary, the marked modulation of striatal activity preceding trial initiation probably reflects subjective changes in reward anticipation as waiting progressed.

## Trial initiation timing modulates striatal activity at predicted outcome time

Striatal activity can be modulated by reward-predictive sensory cues (*Schultz, 2015*). In the present task, a small population of neurons whose activity was modulated at the time of the visual cue (*n* = 27, 3% of all recorded neurons) showed lower activity during premature trials, when the visual cue was not presented $q_{(26)}$ = 6.64, p = 2.15 × 10$^{-4}$ versus timely trials, Tukey post hoc test after significant one-way RM ANOVA ($F_{(1.75, 45.49)}$ = 17.80, p = 5.00 × 10$^{-6}$) (*Figure 5—figure supplement 1*). This modulation by trial initiation timing was similar to that observed in port entry neurons and may represent the same kind of wait time-based reward anticipation activity that extends until when sensory feedback discloses if reward could be obtained or not. In contrast, neurons showing increased activity during licking did

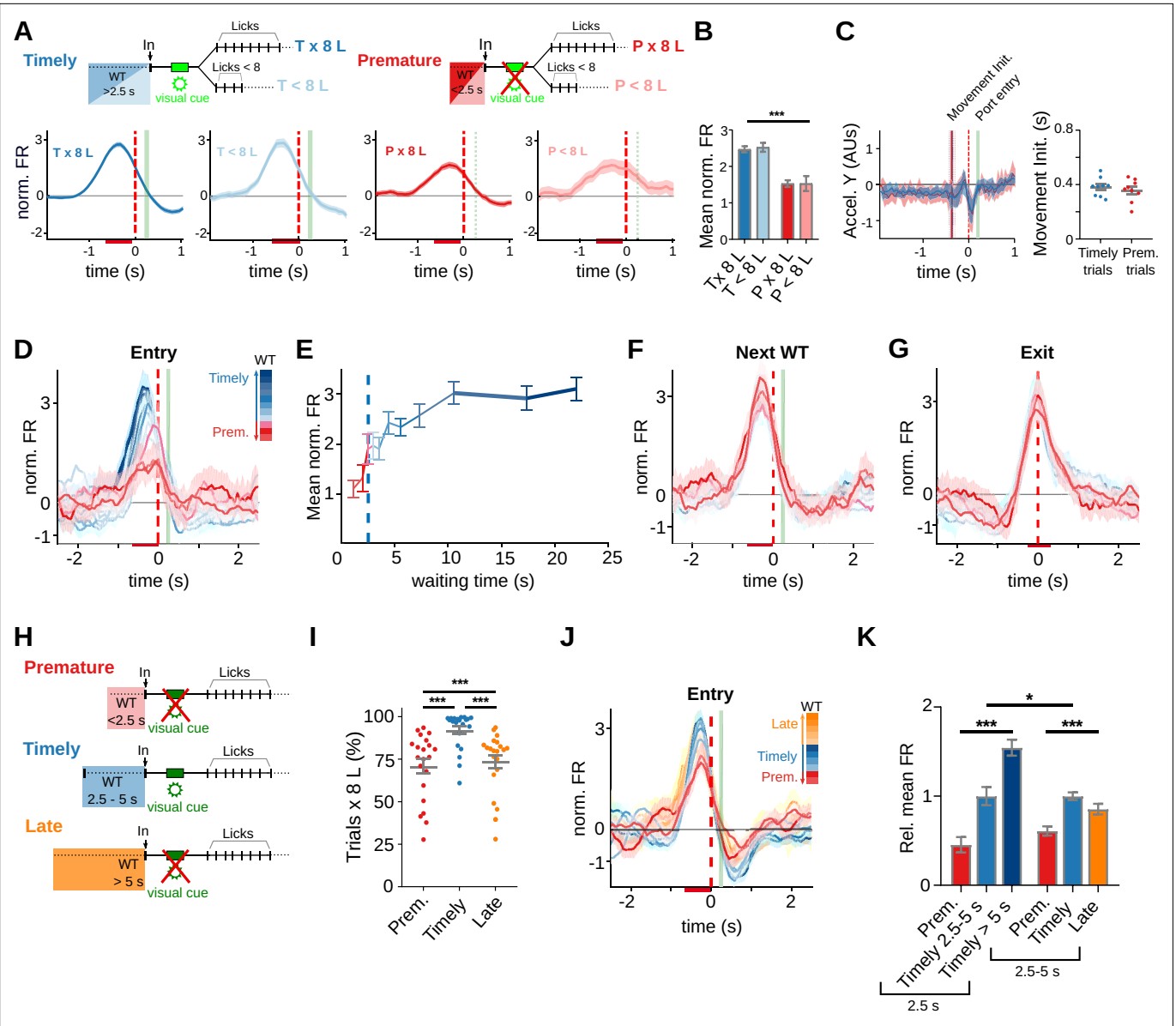

**Figure 4.** Prematurely initiated trials are preceded by low striatal anticipatory activity. Data come from five adult animals performing the standard 2.5-s waiting time task except for (**H–J**), where they correspond to three rats trained in the 2.5- to 5-s cutoff task. (**A**) Average peri-event time histogram (PETH) of entry-related neurons, during premature and timely trials, for trials with or without an 8-lick sequence (diagram on top shows the types of trials analyzed). (**B**) Mean striatal activity during −640 to −80 ms before port entry for all PETH shown in (**A**), ***p<0.001. (**C**) Accelerometer recordings of head movements around port entry time for premature and timely trials. On the left, data are the mean (solid lines) and 95% CI (shaded area) of the accelerometer y-axis. Vertical lines represent the mean (solid) and 95% CI (dashed) of movement initiation for timely and premature trials. On the right, time from port entry to movement initiation for timely and premature trials. Data were obtained from a total of nine training sessions of two animals. (**D**) Average PETH of striatal neurons showing entry-related activity sorted by trial waiting time duration. The color code for the intervals is shown on the right. (**E**) Mean normalized firing rate for each of the waiting time segments. The 2.5-s criterion time is shown with a blue dashed line. (**F**) Average PETH of the same striatal neurons segmented according to next trial waiting time. (**G**) Average PETH of striatal neurons showing activation at port exit segmented as in (**D**). (**H**) Schematic representation of the different trial types in a modified version of the task incorporating a 5-s cutoff time. Timely trials: waiting time between 2.5 and 5 s, Premature trials: waiting time <2.5 s, Late trials: waiting time >5 s. (**I**) Proportion of trials followed by 8-lick sequences for each type of trial n = 21 late sessions, ***p<0.001. (**J**) Average PETH of striatal neurons showing activity preceding port entry according to the waiting time duration. The color code for the intervals is shown on the right. (**K**) Mean normalized firing rate for each of the waiting time groups for the 2.5-s (left) and 2.5- to 5-s (right) waiting times tasks, relative to the mean firing rate of timely trials ***p<0.001; *p<0.05.

The online version of this article includes the following figure supplement(s) for figure 4:

**Figure supplement 1.** Anticipatory activity in the 5-s waiting time task.

not show any modulation by trial initiation timing (*Figure 5A, B*). Furthermore, when we studied the neuronal activity around the individual licks, we found largely overlapping populations of positively modulated neurons with broad activity peaks encompassing many inter-lick intervals (*Figure 5B*). The lick-related modulation lasted longer during rewarded trials, where licking also persisted for longer, than in timely unrewarded (>2.5 s of waiting) and premature 8-lick trials (<2.5 s of waiting). However, it showed a similar amplitude irrespective of reward delivery or omission (*Figure 5B*, activity centered at the eighth lick), suggesting that these neurons were modulated by licking and not by the delivery of the reward itself.

Neuronal activity in the dorsal and ventral striatum is also modulated by the trial outcome and distinguishes between rewarded and non-rewarded trials in probabilistic tasks (*Atallah et al., 2014*; *Histed et al., 2009*; *Nonomura et al., 2018*; *Shin et al., 2018*; *Yamada et al., 2011*). In the present task, there are two types of non-rewarded, yet correctly performed trials: (1) the premature 8-lick trials, which seem to be necessary to maintain an internal representation of the criterion waiting time, but should be minimized to improve the reward rate; and (2) the 50% of timely unrewarded trials that still need to be maintained to sustain the reward rate. Importantly, the animal cannot predict if a timely trial will be rewarded or not but they should be able to predict that no reward will be received after initiating a premature trial as the absence of the light cue unequivocally indicates that no reward will be delivered (*Figure 1B*). To determine if the negative outcome associated with premature 8-lick trials is reported by striatal neurons, and particularly, if this report differs from the report of an unrewarded outcome during timely trials, we looked for outcome-related activity modulations at the time of the eight lick during timely rewarded, timely unrewarded and premature trials with 8-lick sequences (*Figure 5C–G*). Overall, ~15% of the neurons recorded in adult rats (125 out of 867) showed responses at the time of the 8-lick that differed between these trial types. We found neurons that responded preferentially to: (1) reward delivery (*Figure 5C*, 'reward-responsive neurons'), (2) to non-rewarded conditions (*Figure 5D*, 'no reward-responsive'), and (3) to the absence of reward during premature 8-lick trials (*Figure 5E*, 'expected no reward-responsive'). More specifically, 5% of all recorded neurons ($n$ = 49) showed a higher activity modulation when a reward was obtained than in either no-reward conditions, *Figure 5F* top: reward-responsive neurons, Rewarded versus Unrewarded trials, $q_{(48)}$ = 11.18, p = 5.77 × 10⁻¹⁵, Rewarded versus Premature trials, $q_{(48)}$ = 12.52, p = 9.82 × 10⁻¹⁷, Unrewarded versus Premature trials, $q_{(48)}$ = 4.89, p = 1.18 × 10⁻⁵, Tukey test after significant one-way RM ANOVA ($F_{(1.27, 61.14)}$ = 66.01, p = 2.69 × 10⁻¹¹). This difference is observed just before lick rates began to diverge by trial type (*Figure 5H*). Five percent of the striatal neurons showed activation at the time of the no-reward outcome in timely unrewarded trials and were unresponsive to reward delivery (*Figure 5D*). In addition, these neurons showed a marked activation at the time of the eighth lick during premature 8-lick trials (*Figure 5F* middle: no reward-responsive neurons, Rewarded versus Unrewarded trials, $q_{(45)}$ = 12.61, p = 2.23 × 10⁻¹⁶, Rewarded versus Premature trials $q_{(45)}$ = 9.77, p = 1.08 × 10⁻¹², Unrewarded versus Premature trials $q_{(45)}$ = 3.85, p = 3.69 × 10⁻⁴, Tukey test after significant one-way RM ANOVA ($F_{(1.28, 57.7)}$ = 57.11, p = 3.75 × 10⁻¹⁰)). Strikingly, there were 33 neurons (4% of all recorded neurons) that showed a twofold higher activation at the time of the eighth lick in premature trials than in timely unrewarded trials (*Figure 5F* bottom: Rewarded versus Unrewarded trials $q_{(30)}$ = 5.51, p = 5.39 × 10⁻⁶, Rewarded versus Premature trials $q_{(30)}$ = 12.07, p = 4.81 × 10⁻¹³, Unrewarded versus Premature trials $q_{(30)}$ = 10.72, p = 8.86 × 10⁻¹², Tukey test after significant one-way RM ANOVA ($F_{(1.71, 51.39)}$ = 51.34, p = 2.99 × 10⁻⁹)). This difference cannot be explained by differences in lick rates because licking decreased at similar rates after the eighth lick in both types of non-rewarded trials (*Figure 5H*, bottom). Importantly, the no-reward outcome is certain in premature trials as indicated by the absence of the light cue. Further data analysis showed that, indeed, the three populations of outcome-responsive neurons (reward-responsive, no reward-responsive, and 'expected no-reward'-responsive) distinguished between those trials when the timing was right but there was no reward from those where the animal entered the nose-poke prematurely, which are never rewarded. However, those neurons showing a higher activation during premature trials discriminated better between the two no-reward conditions (Rewarded vs. Unrewarded trials $q_{(45)}$ = 0.39, p = 0.694; Rewarded vs. Premature trials $q_{(30)}$ = 5.01, p = 2.23 × 10⁻⁵; Unrewarded vs. Premature trials $q_{(30)}$ = 5.43, p = 6.89 × 10⁻⁶; mixed-effects analysis after significant one-way RM ANOVA ($F_{(1.62, 99.89)}$ = 11.80, p = 8.65 × 10⁻⁴), *Figure 5G*). Overall, the data support that activity at the time of expected trial outcome reflects differences in reward prediction between premature and timely trials.

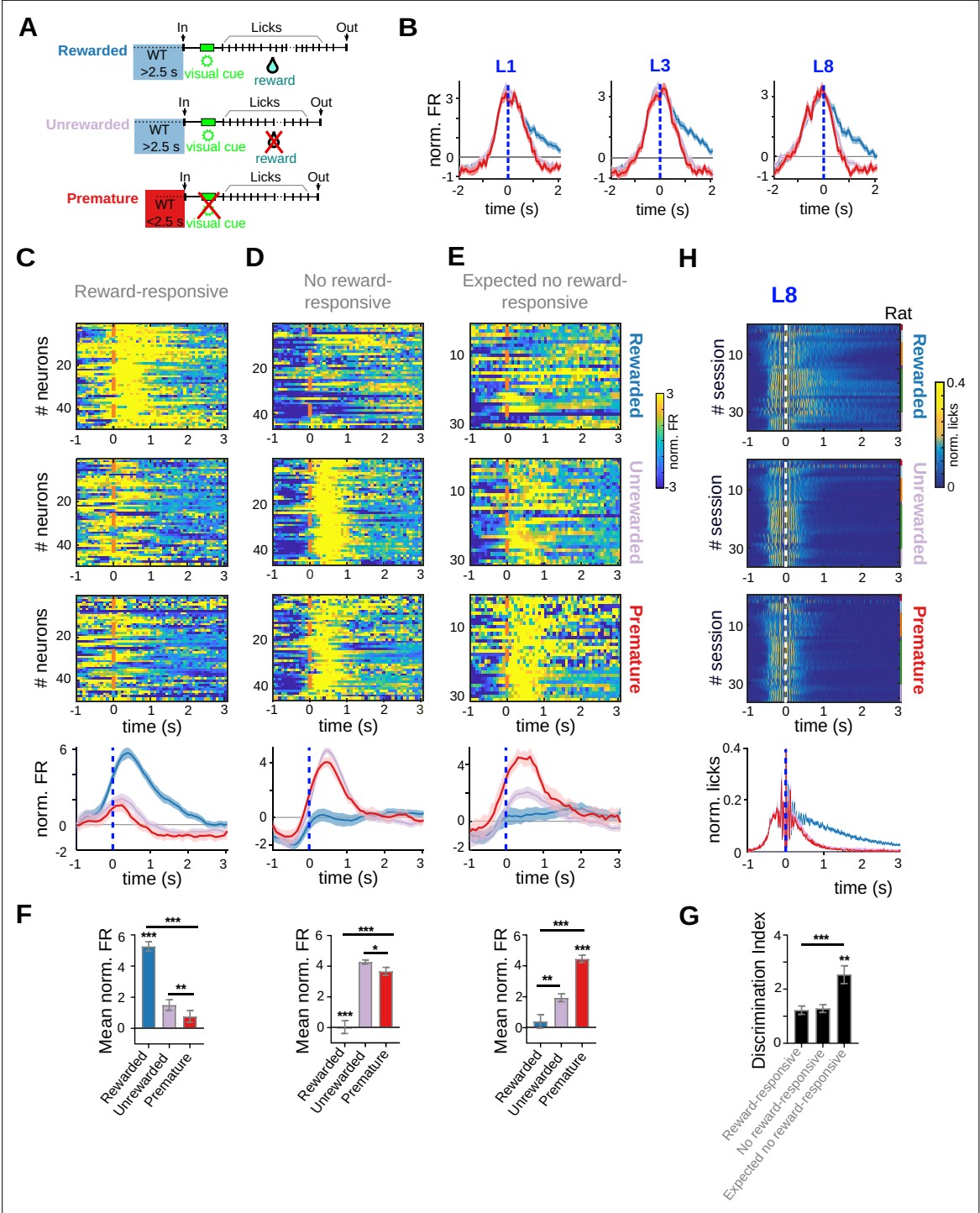

**Figure 5.** Reward-responsive neurons discriminate prematurely from timely initiated trials. Data come from five adult animals performing the standard 2.5-s waiting time task and two adult animals performing the 2.5- to 5-s cutoff version of the task. (**A**) Schematic representation of the different types of trials analyzed. (**B**) Mean (solid lines) and standard error of the mean (SEM) of the average peri-event time histogram (PETH) of striatal units showing firing rate modulations related to licking activity (centered to the first, third, and eighth lick). PETH for all individual neurons and population average PETH of striatal units, aligned to the eighth lick (time 0 s), showing positive firing rate modulations during reward delivery (reward-responsive neurons) (**C**), reward omission during timely trials (no reward-responsive neurons) (**D**), or expected reward omission during 8-lick premature trials (expected no reward-responsive neurons) (**E**). (**F**) Mean normalized firing rate for the different trial conditions (timely rewarded, timely unrewarded, premature) for the three types of neurons depicted in (**C–E**). From top to bottom: reward-responsive neurons, no-reward-responsive neurons, and expected no reward-responsive neurons, ***p< 0.001; **p<0.001, *p<0.05. (**G**) Discrimination index (DI = ABS normalized firing rate in timely unrewarded trials − normalized

*Figure 5 continued on next page*

Figure 5 continued

firing rate in premature trials) for each of the groups of neurons shown in (**C–E**), \*\*\*p< 0.001; \*\*p<0.001. (**H**) PETH of the licking activity for the three trial conditions, centered at the eighth lick, with its corresponding average at the bottom.

The online version of this article includes the following figure supplement(s) for figure 5:

**Figure supplement 1.** Neuronal activity at the moment of the visual cue.

## Activity at port exit reports whether the animal has performed the learned action sequence or not

Although striatal activity predominated at the boundaries of the learned behavioral response, the activity preceding response initiation was not invariably connected to the execution of the learned action sequence. However, when all port exit neurons were considered (i.e., port exit only plus port entry/port exit neurons), a higher activity was observed for 8-lick trials than for trials with an incomplete licking sequence, independently of whether the trials were timely initiated or not (**Figure 6A, B**; $F_{(1, 101)} = 36.28$, p = $2.79 \times 10^{-8}$ for lick sequence structure, no effect of trial timing $F_{(1, 101)} = 1.86$, p = 0.176, interaction $F_{(1, 101)} = 4.07$, p = 0.046, two-way RM ANOVA). Moreover, there was no difference between trial types in the accelerometer data at the time animals exited the port (**Figure 6C**). Thus, port exit-related activity seems to tell if the learned response was emitted or not regardless of its timing and outcome.

## Adolescent rats make more premature trials

Adolescence is a high impulsivity period more prone to make high-risk decisions, while neurodevelopmental processes are still taking place (**Romer, 2010**). In order to better understand how adolescent animals behave in this self-initiated task where an internal representation of time should be kept to withhold a prepotent response, we recorded striatal activity from five 35- to 46-day-old rats and trained them in the 2.5-s waiting interval task. The rats successfully improved their performance in the task as shown by a higher percentage of correct trials (**Figure 7A**), a higher reward rate (**Figure 7B**, p = $2.00 \times 10^{-6}$, Wilcoxon matched pairs signed-rank test *n* pairs = 20), and a faster performance with training (**Figure 7—figure supplement 1**). Trial duration, latency to first lick and 8-lick sequence duration did not differ between 8-lick trials of adolescent and adult rats (**Figure 7—figure supplement 1A–C**), nor between premature 8-lick and timely 8-lick trials in adolescent rats (**Figure 7—figure supplement 1E–G**). Accelerometer recordings of head movements available from two adolescent rats did not show differences of movement initiation time between premature and timely 8-lick trials (0.591 ± 0.014 and 0.637 ± 0.022 mean and SEM – seconds for premature and timely trials, respectively, $t_{(9)}$ = 1.537, p = 0.1586, Paired *t*-test; **Figure 7—figure supplement 1I**). Finally, the proportion of premature 8-lick sequence trials increased with training in adolescent rats (**Figure 7C**; significant effects of

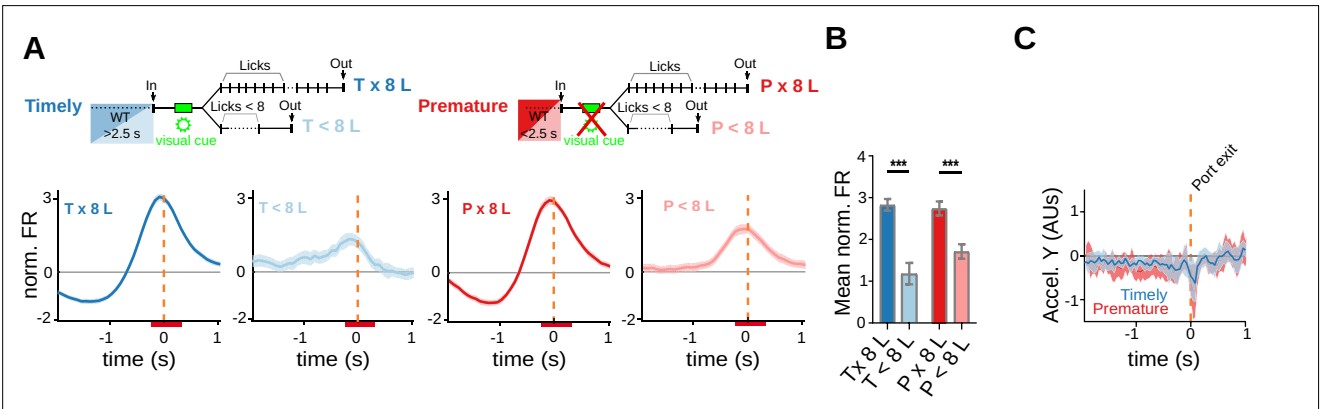

**Figure 6.** Striatal activity at port exit reports the performance of the action sequence. (**A, B**) Data come from five adult animals performing the standard 2.5-s waiting time task. (**A**) Average peri-event time histogram (PETH) of neurons responding to port exit, during premature and timely trials, for 8-lick and <8-lick sequence trials (diagram on top shows the types of trials analyzed). (**B**) Average striatal activity corresponding to the PETH shown in (A). Colored bars over the *x*-axis show the interval used to detect firing rate modulations, \*\*\*p<0.001. (**C**) Mean (solid lines) and 95% CI (shaded area) of head acceleration around port exit for premature and timely trials. Data were obtained from a total of nine training sessions of two animals.

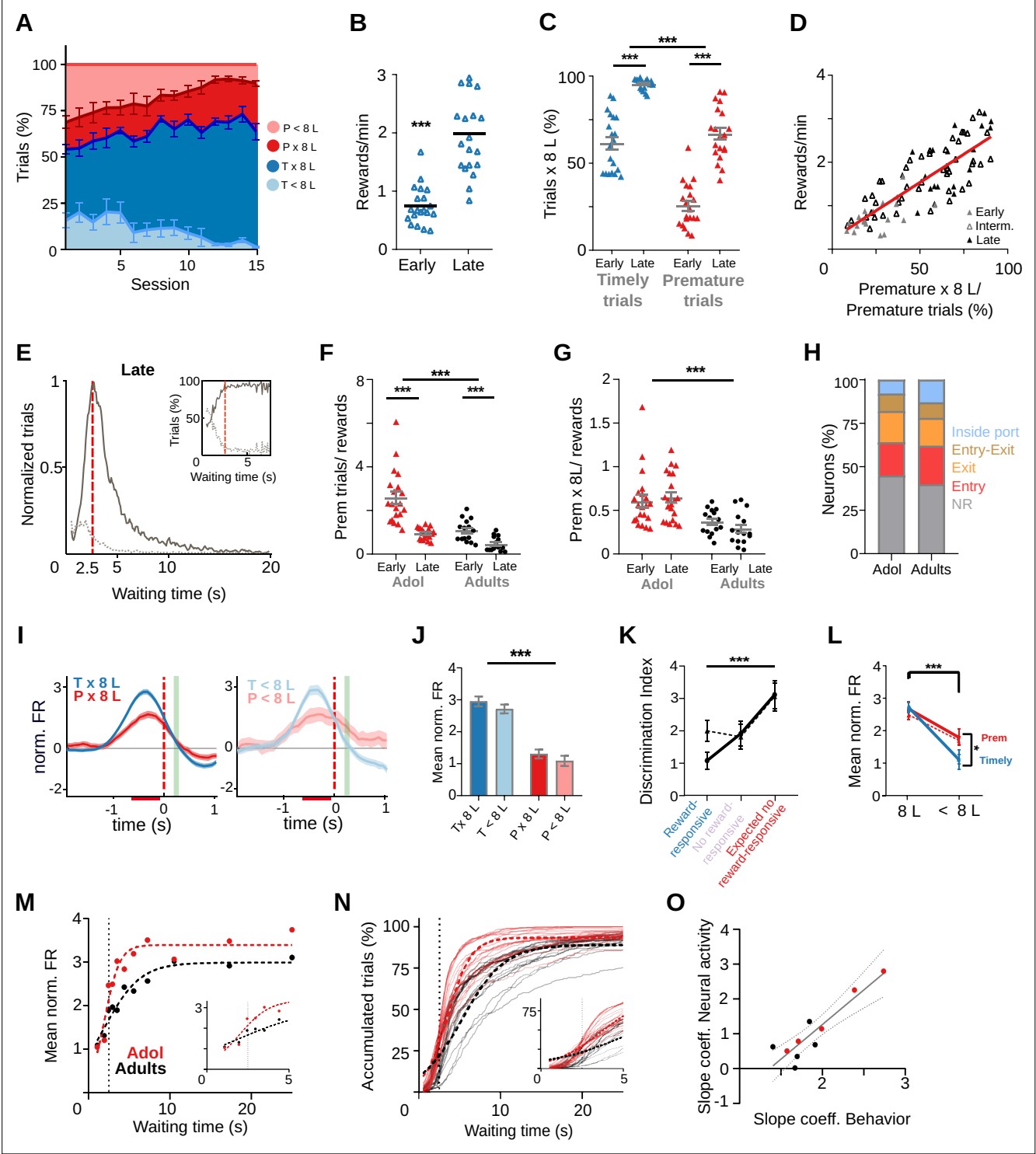

**Figure 7.** Adolescent rats are more impulsive and show a steeper wait time modulation of striatal activity. Data come from six adolescent and five adult rats performing the standard 2.5-s waiting time task. (**A–E**) Behavior of adolescent rats in the 2.5-s waiting time task. (**A**) Percentage of the different trial types per session. (**B**) Reward rates for each training stage. (**C**) Proportion of trials followed by 8-lick sequences at early and late learning stages. (B, C) Data are expressed as mean ± standard error of the mean (SEM), $n = 20$ early and 20 late sessions, *** $p<0.001$; *$p<0.05$. (**D**) Correlation between percentage of prematurely initiated trials followed by an 8-lick sequence and reward rate ($Y = 0.026*X + 0.112$, Slope significantly different from zero, $p = 5.88 \times 10^{-22}$, n=86 sessions from 6 rats). (**E**) Normalized frequency distributions of trial initiation times (waiting time), separated for trials with (dark gray) and without (light gray dotted line) 8-lick sequences, in late stages of training ($n = 20$ sessions). Inset: Percentage of trials with and without 8-lick sequences for each bin, zoomed around the criterion time. (**F**) Ratio between the number of premature trials and obtained rewards for adolescent

*Figure 7 continued on next page*

*Figure 7 continued*

and adult rats, ***p<0.001. (**G**) Proportion of premature trials with 8-lick sequence per obtained rewards for adolescent and adult rats, ***p<0.001. (**H**) Percentage of neurons showing task-related firing rate modulations in adolescent rats. (**I**) Average peri-event time histogram (PETH) of entry-related neurons, during premature and timely trials, for trials with (x8L) or without an 8-lick sequence (<8L), in adolescent rats. (**J**) Average striatal activity corresponding to the PETH shown in (**I**), ***p<0.001. (**K**) Outcome-related neuronal activity: Discrimination index between 8-lick timely unrewarded and premature trials for different types of outcome responsive neurons from adolescent and adult rats. (**L**) Port exit-related activity: mean normalized firing rate at the time of port exit for trials with or without 8-lick sequence for both ages. (**K, L**) Dashed lines correspond to adolescents and full lines to adults,***p<0.001; *p<0.05. (**M**) Waiting time modulation of anticipatory activity: mean normalized firing rate for each of the waiting time segments for each age group. Dashed lines show the logistic function fitted to the binned neuronal activity data. Inset: Detail around 2.5-s waiting time. (**N**) Cumulative frequency distribution of 8-lick trial initiation times for adolescent and adult rats. Each light color line shows an individual session. Dashed lines show the logistic function fitted to the behavioral data. Inset: Detail around 2.5-s waiting time. (**O**) Correlation between p2 slope coefficients of the logistic functions fitting the behavioral and neurophysiological data of each rat.

The online version of this article includes the following figure supplement(s) for figure 7:

**Figure supplement 1.** Adolescent rats make more premature trials.

**Figure supplement 2.** Striatal activity also marks transitions between behavioral states of the task in adolescent animals.

**Figure supplement 3.** Anticipatory, reward-responsive and exit activity in striatal neurons of adolescent rats.

**Figure supplement 4.** Behavioral performance of non-implanted adolescent and adult rats.

learning stage ($F_{(1, 19)}$ = 93.97, p = 8.66 × 10$^{-9}$) and trial timing ($F_{(1, 19)}$ = 211.67, p = 9.40 × 10$^{-12}$), no interaction ($F_{(1, 19)}$ = 5.01, p = 0.037), two-way RM ANOVA), in parallel with reward rate (*Figure 7D*, $R^2$ = 0.067, $F_{(1, 84)}$ = 170.98, p = 5.88 × 10$^{-22}$, n = 86 sessions from 6 rats), and the frequency distribution of trial initiation times showed that adolescent rats learned the waiting interval (*Figure 7E*). In summary, adolescent rats also released the learned action sequence prematurely despite extensive training, and indeed, they did it more frequently than adult rats (*Figure 7F, G*). The number of premature trials per reward obtained was twofold higher in adolescent than adult rats even after training (*Figure 7F*; significant effects of age group ($F_{(1, 34)}$ = 28.51, p = 6.22 × 10$^{-6}$), training ($F_{(1, 34)}$ = 50.19, p = 3.49 × 10$^{-8}$), and interaction ($F_{(1, 34)}$ = 10.43, p = 2.74 × 10$^{-3}$), two-way RM ANOVA), and this was due to preferential retention of the premature 8-lick trials through training (*Figure 7G*; significant effect of age group ($F_{(1, 34)}$ = 17.88, p = 1.67 × 10$^{-4}$), no effect of training ($F_{(1, 34)}$ = 0.19, p = 0.662), no interaction ($F_{(1, 34)}$ = 1.58, p = 0.217), two-way RM ANOVA). Altogether, adolescent rats achieved the same reward rate as adult rats but at the expense of a higher cost as evidenced by the excess of premature responses per reward obtained.

## Steepest reward anticipation signal preceding trial initiation in adolescent rats

Overall, task-related striatal activity was qualitatively similar in adolescent and adult rats (*Figure 7—figure supplement 2*). Of the registered units in adolescent rats trained in the 2.5-s waiting interval task (n = 876), the proportion of neurons showing port entry-, port exit-, and lick-related activity was similar to that found in adult rats (*Figure 7H*; chi-square$_{(4)}$ = 1.87, p = 0.759, chi-square test). Also like in adult rats, the activations preceding port entry were higher for timely than for premature trials regardless of whether the 8-lick sequence was completed or not (*Figure 7I, J*; significant main effect of trial initiation timing $F_{(1, 362)}$ = 148, p = 3.72 × 10$^{-27}$, no effect of action sequence structure, $F_{(1, 362)}$ = 1.04, p = 0.309, no interaction $F_{(1, 362)}$ = 0.12, p = 0.723, Two-way RM ANOVA). Outcome-related activations discriminating reward delivery from reward omission, and selective activations at expected reward time during premature 8-lick trials, not attributable to differences in lick rate, were also present (*Figure 7—figure supplement 3*).

Therefore, we looked for quantitative differences in task-related activities that could account for the more impulsive behavior of adolescent rats. Data from six adolescent and five adult rats trained in the 2.5-s waiting interval task were used for the following analysis. Activations at the time of expected reward (*Figure 7K*, effect on trial initiation timing ($F_{(1.95, 76.18)}$ = 8.53, p = 4.64 × 10$^{-3}$), no effect of age ($F_{(1, 50)}$ = 0.42, p = 0.518), no interaction ($F_{(2, 78)}$ = 0.95, p = 0.390), Restricted Maximum Likelihood [REML] test) and activity at port exit – probably reflecting that the correct action sequence was emitted (*Figure 7L*, sequence structure, $F_{(1, 356)}$ = 124.14, p = 6.18 × 10$^{-25}$, timing $F_{(1, 356)}$ = 4.11, p = 0.043, significant interaction between trial initiation timing and sequence structure $F_{(1, 356)}$ = 21.21, p

= 5.74 × 10$^{-6}$, three-way ANOVA), did not differ between adolescent and adult rats (effect of age $F_{(1, 356)}$ = 0.12, p = 0.727).

On the other hand, the data displaying the amplitude modulation of striatal activity preceding trial initiations at increasing waiting times (n = 235 and 124 striatal neurons for adolescent and adult rats, respectively) differed between adolescent and adult rats (*Figure 7M*). To substantiate the dynamics of the age-related differential modulation of anticipatory activity, we adjusted a logistic function to the binned neuronal activity data of each age group (*Figure 7M*). Despite the relatively low number of 8-lick premature trials, which forced us to use relatively long bins for the shortest waiting times, the equation fitted the data convincingly ($R^2$ = 0.904 and 0.945 for adolescent and adult rats, respectively). Importantly, the adolescent rats curve showed a significantly steeper increase than the adult rats curve (p2 coefficient of the logistic function, profile likelihood estimation of the 95% confidence interval: adolescent rats = 1.080 (95% CI: 0.533–2.160) and adult rats = 0.4525 (95% CI: 0.273–0.747); $F_{(1, 16)}$ = 4.86, p = 0.042). Interestingly, the difference between the curves was more evident during the first seconds that follow the criterion wait time than for shorter waiting times (*Figure 7M*, inset). To understand the behavioral correlate of the observed neuronal activity differences, the same logistic function previously used to fit the neuronal data was adjusted to the cumulative frequency distribution of 8-lick trial initiation times of each age group (*Figure 7N*, $R^2$ = 0.9307 and 0.8874, for adolescent and adult rats, respectively). As expected, the curve of the adolescent group showed a significantly steeper growth rate (p2 coefficient of the logistic function, profile likelihood estimation of 95% confidence interval: adolescent rats = 0.703 (95% CI: 0.683–0.724) and adult rats = 0.392 (95% CI: 0.380–0.406), $F_{(1, 9960)}$=658.8, p = 9.60 × 10$^{-14}$). Similar behavioral results were obtained in an independent sample of 6 adolescent and 6 adult rats without electrode implants (*Figure 7—figure supplement 4*). Remarkably, in addition to a relative excess of premature 8-lick trials, the adolescent rats showed relatively more timely trials than adult rats at short times after the criterion wait time was surpassed (*Figure 7N*), suggesting that adult rats are more tolerant to reward delay. Indeed, for a waiting time of 5.1 s, adult rats had accumulated 50% of 8-lick trials, while adolescent rats had already accumulated 70% of the trials. Finally, we adjusted the logistic function to the data of each rat separately. We then computed a correlation between the p2 parameters obtained for the waiting modulation of neuronal activity and trial initiation times, and found a significant correlation (*Figure 7O*, $R^2$ = 0.833, p = 2 × 10$^{-4}$, n = 10).

In sum, the data show that the waiting modulation of reward anticipation grows at a faster rate in adolescent rats, paralleling a higher rate of accumulation of initiated trials across time in this age group.

## Discussion

In the present task, water-deprived rats learn to withhold a prepotent response to a water port to avoid a negative contingency, the re-initiation of the waiting interval, which delays the opportunity to get the next reward. The task resembles differential reinforcement of low rates of responding protocols (DRL) that are used to assess response inhibition and timing factors associated with impulsivity (*Monterosso and Ainslie, 1999*; *Neill, 1976*). Factors that increase premature responding in humans, such as psychostimulant drugs (*Evenden, 1998*), psychostimulant withdrawal (*Peterson et al., 2003*), D2-type receptor agonists (*Engeln et al., 2016*), maternal separation and social isolation (*Lovic et al., 2011*), sleep restriction (*Kamphuis et al., 2017*), and adolescence (*Andrzejewski et al., 2011*) also increase premature responding in rodents trained with DRL procedures. Furthermore, these procedures have been used in educational and clinically meaningful contexts to reduce impulsive behavior (*Bonner and Borrero, 2018*; *Lennox et al., 1987*). Unlike classical DRL procedures, the present task requires responding with an action sequence that becomes highly automatized with training and could not always be stopped despite sensory feedback telling that it will not be rewarded. Interestingly, although premature responding showed an overall decrease with training, the premature releases of the learned action sequence were selectively preserved, suggesting that these responses are necessary to learn the duration of the waiting interval and to adapt to its changes. As long as the waiting interval remains predictable, investigation of the waiting interval through premature responding should be minimized to improve reward rate. Importantly, adolescent rats learn the task but make more premature responses per reward obtained. In addition, they initiate their timely trials closer to the criterion wait time than the adult rats, supporting that they are less patient. Overall, the

adolescent rats seem to take a higher risk of releasing the learned action sequence prematurely, which makes them more impulsive than adult rats in the present task.

Recent studies link premature responding in anticipation of reinforcement to a higher tendency to automatize behavior and form habits (*Everitt et al., 2008*; *Voon et al., 2015*). Striatal activity marking the boundaries of automatized action chains has been perceived as a sign of 'packaged behavioral sequences' (*Graybiel, 2008*) that would be difficult to stop after their release regardless of whether their initiation was more or less goal directed (*Geddes et al., 2018*; *Robbins and Costa, 2017*). While this 'bracketing activity' prevails in the DLS, according to some studies, the representation of task events contributing to goal-directed behavior persists in the DMS even after extensive training (*Kubota et al., 2009*; *Thorn et al., 2010*). These parallel representations may allow switching between automatic and deliberation-based task-solving strategies when outcomes change (*Balleine and Dickinson, 1998*; *Daw et al., 2005*; *Yin and Knowlton, 2006*) or interventions that modify neural activity impair behavioral control by one of the circuits involved (*Gremel and Costa, 2013*; *Smith and Graybiel, 2013*; *Yin et al., 2004*). The remarkably similar behavior observed in 8-lick prematurely released and timely unrewarded trials led us to expect that a stronger boundary activity, and/or a less precise coding of task events, could explain the higher rates of premature responding observed in adolescent rats. Indeed, a recent study suggested that the stronger the activity at the action initiation, the lower the deliberation at the turn-choice site in a T-maze task (*Smith and Graybiel, 2013*). Here, the strength of anticipatory activity increased with the time waited before response release and was higher in the more impulsive adolescent rats. However, it did not predict that the rewarded 8-lick sequence would be included in the behavioral response. On the other hand, a similar activity followed all responses containing the learned 8-lick sequence regardless of trial outcome and despite the markedly different lick rates observed between rewarded and unrewarded responses after outcome disclosure. A recent study showed that hierarchical control during action sequence execution may allow the selective removal of intermediate sequence elements (*Geddes et al., 2018*), which could be a mechanism through which licks could be deleted from behavioral responses preceded by similar wait time-based reward expectations in the present task. Moreover, since our recordings were mainly obtained from the medial and central dorsal striatal regions, we cannot rule out that a different anticipatory activity specific for the learned sequence emerges in the DLS in our task as previously reported for other tasks (*Jin and Costa, 2010*; *Jog et al., 1999*; *Martiros et al., 2018*; *Thorn et al., 2010*). Importantly, the activity of many striatal neurons seemed to continuously follow the expression of the lick sequence as has also been reported previously by others (*Jin and Costa, 2010*; *Jin et al., 2014*; *Rueda-Orozco and Robbe, 2015*) and a recent study showed a continuous modulation of striatal neuronal activity across the initiation, execution, and termination of cue-guided locomotion (*Sales-Carbonell et al., 2018*). Overall, our data show that the concentration of striatal activity at the beginning and end of a learned task is widely expressed in the dorsal striatum even after skilled performance is reached, as reported in a recent study with a different task (*Vandaele et al., 2019*), and suggest that, this activity is more flexible regarding the properties of the behavioral response it bounds, and carries information about its pertinent timing.

We speculated that the more impulsive behavior of adolescent rats observed in the present task could relate to changes in wait time-based modulation of reward anticipation and/or outcome evaluation signals. Striatal signals at the time of expected outcome not only discriminated between rewarded and unrewarded correct trials as observed by others (*Atallah et al., 2014*; *Nonomura et al., 2018*; *Shin et al., 2018*), but also premature from timely responses. A specific outcome evaluation signal after premature 8-lick trials may serve to inhibit premature responding and minimize the exploration of the waiting interval. To compute such outcome signal, the animal has to retain information regarding the time waited before responding (or regarding visual feedback on proper trial initiation timing) while monitoring the execution of the lick sequence. Such kind of integration could be implemented by retaining information of early task events along the sequential activation of striatal ensembles (*Her et al., 2016*; *Nonomura et al., 2018*). However, although the effect of waiting time on task-related striatal activity extended until the time of the visual cue, it was absent in ensembles with lick-related activity. Thus, the sustained lick-related signal may serve to estimate outcome timing (*Zold and Hussain Shuler, 2015*), but information about trial initiation timing or the visual cue should arrive at outcome-sensitive ensembles through other mechanisms likely involving inputs from prefrontal and orbitofrontal cortex areas (*Asaad et al., 2017*; *Hamid et al., 2021*; *Wassum et al., 2011*). Nonetheless,

we found no differences in outcome evaluation signals that could explain the more impulsive behavior of adolescent rats. On the other hand, although premature responding has classically been linked to poor behavioral inhibition (*Bari and Robbins, 2013*), alternative views equates it to preferring a smaller more immediate reward over a larger delayed one (*Dalley and Robbins, 2017*; *Monterosso and Ainslie, 1999*), and to deficient gathering or evaluation of evidence while waiting (*Dalley and Robbins, 2017*). Interestingly, influential models propose that time estimates derived from accumulation of pacemaker counts can be compared to a memorized time interval to decide whether a target time has been reached or not (*Buhusi and Meck, 2005*; *Gibbon, 1977*; *Namboodiri and Hussain Shuler, 2016*). Previous studies showed that dorsal striatum ensembles can provide time estimations (*Bakhurin et al., 2017*; *Emmons et al., 2017*; *Gouvêa et al., 2015*; *Matell et al., 2003*; *Mello et al., 2015*; *Zhou et al., 2020*) and we also found sequential and tonic ensemble activation during the waiting period (*Figure 3B* and *Figure 3—figure supplement 1A*) that could serve to track time while waiting in the present task. By comparing the time accumulated while waiting against the memorized waiting interval, the animal could anticipate how likely is the upcoming action to be rewarded. How fast reward expectancy steps up when time waited approaches the reference time interval would depend on several factors, including temporal discounting effects on reward value (*Monterosso and Ainslie, 1999*; *Namboodiri and Hussain Shuler, 2016*; *Wittmann and Paulus, 2008*). In this sense, the anticipatory signal we observe could be interpreted as a reading of the temporal discount function at the time chosen to release the learned action sequence. A recent theory proposes that animals time their decisions by estimating if they can improve the reward rate experienced in the recent past; according to it, the longer the time window over which reinforcement history is estimated, the higher the tolerance to delays of future rewards (i.e., the less steep the temporal discount effect) (*Namboodiri et al., 2014*). Adults should be able to integrate information about past reinforcement history over longer time windows that adolescents (*Namboodiri et al., 2014*), which is consistent with the finding that temporal discount rate decreases throughout childhood and adolescence (*Green et al., 1999*; *Scheres et al., 2006*; *Steinberg et al., 2009*). Alternative theories propose that waiting impulsivity relates to perceiving durations longer than they are, which would be associated to perceiving a higher cost of time and with a steeper temporal discounting (*Wittmann and Paulus, 2008*). Moreover, time perception depends on several cognitive processes that are modified by adolescence including working memory, attention, and mood (*Baumann and Odum, 2012*; *Wittmann and Paulus, 2008*). It has been also been proposed that impulsivity relates to changes in an internal urgency signal that influences the timing of decisions and may be primary responsible for the build-up of neural activity observed in the striatum and cortex that often precedes action initiation (*Carland et al., 2019*).

Interestingly, in the present task, adolescent rats achieve a similar reward rate to adult rats and, indeed, they do not show the negative effect of premature trials on reward rate that is observed in adults (*Figure 7—figure supplement 1*). Importantly, although they achieve this reward rate at the expense of a higher energy cost (more premature trials per reward obtained), by prematurely executing the learned action sequence, adolescents would likely get more precise information regarding the duration of the waiting interval, which, in turn, would refresh more frequently their cognitive map of the task. This view is consistent with the idea that impulsivity is advantageous when it helps to adapt to uncertain environments and that increased exploratory behavior in adolescence allows gaining knowledge that would be useful to guide decisions in adulthood (*Addicott et al., 2017*; *Spear, 2000*). In this context, the steepness of the wait time-based reward anticipation signal could be under the influence of a gain factor that, according to reinforcement learning models, reflects the degree of preference for the highest value option and regulates explore–exploit tradeoff (*Addicott et al., 2017*). Further studies are needed to disclose which factors influence the wait time-based reward anticipation signal we observe in the present task and which among them are responsible for the differences observed between adult and adolescent rats. Moreover, since the experiments have been conducted in male rats and impulsivity has sexually dimorphic features (*Weafer and de Wit, 2014*), studies are needed to understand how female animals behave in the present task and whether female and male animals encode wait time-based reward anticipation differently.

In summary, detailed analysis of behavior in a waiting task that promotes premature responding with an automatized action sequence shows a more impulsive behavior in adolescent rats. Unlike other tasks previously used to study neural correlates of waiting impulsivity (e.g., the 5-choice serial reaction time task), the present task is learned rapidly, which makes it suitable for studying premature

responding during rodent adolescence. Moreover, by requiring self-paced responding with a stereo-typed action sequence, and making uncertain reward obtention during timely trials, the task generates conditions that 'clamp' behavior. This allowed identifying a modulation of anticipatory striatal activity by time waited before responding, which grows with a steeper slope in adolescent rats, likely reflecting age-related changes in temporal discounting, internal urgency states or explore–exploit balance. Translational studies are necessary to understand if similarly designed tasks capture the relationship between impulsivity and automaticity in behavior that has been related to vulnerability to drug addiction.

## Materials and methods

### Subjects

Subjects were adult male Long-Evans rats (adolescents: 28- to 46-day-old at the beginning of the experiments, weight ~120 g; adults: 70–128 days old at the beginning of the experiments, weight ~390 g), kept under a 12-hr light/dark cycle (lights on at 7:00 A.M.) at a constant temperature range of 21–23°C. Rats were housed in groups of 3–4 in regular cages up to the time of surgery, after which they were singly housed with moderate environmental enrichment (toys, tissue paper strips). Two days before the beginning of the behavioral training, subjects entered a water restriction protocol in which they received free access to water for 20 min each day. Once a week, animals were allowed to have water ad libitum. This schedule maintained animals at 90% of their pre-deprivation weight. All procedures complied with the National Institutes of Health Guide for Care and Use of Laboratory Animals (Publications No. 80-23, revised 1996) and were approved by the Animal Care and Use Committee of the School of Medicine of the University of Buenos Aires – CICUAL – Resolución CD 2344/2015, Resolución CD 3101/2017, RES CS-2020-1053. Striatal recordings were obtained from: five adult males that were part of the 2.5-s WT experiments, three adult males that were part of the 5-s WT experiments (*Figure 2I* and *Figure 2—figure supplement 1*) and another three adult males were included in the 2.5- to 5-s WT experiments (*Figure 4H–K*, *Figure 2—figure supplement 2* – all except S2f, see below). Six adolescent males were part of 2.5-s WT experiments (*Figure 7*, *Figure 7—figure supplements 1–3*). In addition, six adult and six adolescent male rats, not implanted with recording electrodes, were part of the behavioral experiments reported in *Figure 7—figure supplement 4*.

### Electrodes

Eight tetrodes (each made of four 12 µm, tungsten wires, California fine wire company, USA, 0.2 MΩ impedance) were attached to a homemade micromanipulator (*Vandecasteele et al., 2012*). The array of eight tetrodes was gradually lowered each day before training to find neuronal activity. At the end of the recording session the array was also lowered to change the recording site for the next training session (approximately 80 µm each day).

### Surgery

Before surgery, subjects were treated with local anaesthetic (lidocaine) on the scalp. Under deep anesthesia with isoflurane (3–4% for induction, 1–1.5% for maintenance), rats were placed in the stereotaxic frame and chronically implanted with an array of tetrodes aimed at the striatum (AP: −0.12 cm, L: −0.25 cm, DV: −0.35 cm). Body temperature was maintained using a heating pad. Through a small craniotomy performed in the corresponding area the tetrodes were then lowered to a depth of 3.5 mm. The micromanipulator was fixed in place with dental cement. Two stainless steel screws (0–80× 1/8″ Philips Pan head) were inserted above the cerebellum to be used as ground and reference. Three additional steel screws were inserted to anchor the whole implant to the skull. The craniotomy was covered with a sterile 50–50 mixture of mineral oil and paraffin. Toward the end of the surgical procedure, animals were treated with antibiotics (i.m., enrofloxacin 10 mg/kg) and a veterinary ointment (antiseptic, anti-inflammatory, and anesthetic) was applied on the skin in contact with the external side of the implant to prevent microbial infections. Subjects were kept under careful observation until awakening. The first recording session followed 7 days of post-surgical recovery.

## Behavioral training

Rats were placed in a dark behavioral operant chamber that contained a nose-poke where they could seek a water reward by licking through a slot onto a lick tube (Med associates) (*Figure 1A*). After a minimum of 2.5 s (criterion waiting time), rats could initiate a trial by entering the nose-poke, 200 ms after which they received a 100 ms visual cue through two green leds placed on the sides of the nose-poke. The animal was then required to lick eight times to gain a water reward (~20 µl of water) in 50% of the trials in a pseudo-random manner. Licks were detected through an infrared beam located at the sides of the lick tube. Trials ended when the animal removed its head from the nose-poke. Sessions ended when the subject completed 300 trials or when the session reached 120 min. Premature trials are those in which the criterion waiting time was violated (*Figure 1B*). Once the animal reached 80% of timely trials with a complete 8-lick sequence for two consecutive days, sessions were considered 'late sessions'. 'Early' sessions were considered from the first day of training and the number of early sessions was determined to match those classified as 'late' for each animal, for comparison purposes. The remaining sessions between 'early' and 'late' were classified as 'intermediate'.

## Variants of the rewarded task

We performed two additional versions of the task: one in which the criterion waiting time was duplicated (5-s waiting time task) and another in which the animals had to enter the nose-poke within a waiting interval of 2.5–5 s (2.5- to 5-s waiting time task). For the 5-s waiting time task, subjects were trained just like in the 2.5-s task, being the minimum waiting interval of 5 s. After reaching a stable performance with the 5-s waiting interval, two of the animals were switched to the 2.5-s protocol (data shown in *Figure 2I*).

For the 2.5- to 5-s waiting time task, rats were trained with a 'shaping protocol'. Briefly, they were trained for three consecutive sessions with a 2.5-s task and afterwards we incorporated an upper limit to the waiting interval: any trial started with a waiting interval longer than the upper limit was not rewarded. The upper limits for shaping were 15 s (one session), 10 s (two sessions), and 5 s (one to three sessions). One day after the end of shaping, tetrodes were chronically implanted with the surgical procedure described above. Trials in the 2.5- to 5-s waiting time task were classified as premature (if animals entered the nose-poke before the lower limit of 2.5 waiting), timely (if they entered the nose-poke between 2.5 and 5 s of waiting) and late (when they entered the nose-poke after 5 s of waiting). Premature and late trials were not rewarded and timely trials were rewarded with a probability of 0.5 (and reported to the animal with the visual cue).

## Accelerometer recordings

Accelerometer data were obtained from two adult and two adolescent rats. Data from adult rats were collected with a MPU-6050 accelerometer (sampling rate of 25 Hz) and registered using an Arduino UNO board connected to a PC via PLX-DAQ software (Parallax Inc). Accelerometer data from adolescent rats were obtained with an ADXL335 accelerometer (sampling rate of 125 Hz) and registered using an Arduino UNO board and a custom-made script in Processing software (https://processing.org/).

The three-axis acceleration data were then band-pass filtered (0.25–6 Hz cutoff, fourth-order Butterworth filter) after which the Euclidean norm was taken. This acceleration signal was segmented into 10-sample windows without overlapping for the adolescent data and into 10-sample windows with 80% overlapping for the adult data (in both cases the time resolution was of 80 ms). The standard deviation for all data points in each window was calculated. Windows with SD <3.5 were considered as a no-movement window. We limited the analysis to movement initiation times detected up to 1.2 s before port entry. In addition, only trials with a waiting time below 5 s were considered in order to include trials where animals are clearly engaged in the task.

## Histology

Animals were given a lethal dose of ketamine–xilazine and transcardially perfused with cold phosphate-buffered saline (PBS) containing heparin (500 U/l) followed by 4% paraformaldehyde in PBS. The brain was quickly removed, postfixed in PFA at 4°C for 2–12 hr, and placed in a 30% sucrose solution in PBS. Frozen coronal sections (50 µm) were collected with a sliding microtome, and histological verification

of the electrode endpoints and recording tracks was done in microscopy fluorescent pictures. Later, sections were safranin stained to confirm the visualization of the tracks.

## Quantification and statistical analysis

### Neural recording and data analysis

Brain activity and behavioral data were collected during the training sessions using commercially available hardware and software (sampling rate 32.5 kHz, Cheetah, Neuralynx). Neurophysiological and behavioral data were explored using NeuroScope (http://neuroscope.sourceforge.net; *Hazan et al., 2006*). Spike sorting was performed automatically, using KlustaKwik (http://klustawik.sourceforge.net), followed by a manual adjustment of the clusters (using 'Klusters' software package; http://klusters.sourceforge.net, *Hazan et al., 2006*). After the spike sorting procedure, all data were analyzed with MATLAB software using custom-built scripts and 'FMAtoolbox' (http://fmatoolbox.sourceforge.net/). We registered a total of 867 units in adult rats: 244 were task responsive and 158 non-responsive in the 2.5-s waiting time task, 47 task-responsive and 45 non-responsive in the 5-s waiting time task, and 294 task-responsive and 79 non-responsive in the 2.5- to 5-s waiting time experiments. In adolescent rats, we registered a total of 876 units, 385 with no response and 491 task-responsive, always in the standard 2.5-s waiting time task. All subjects and their corresponding experiment and units are detailed in *Table 1*.

PETHs were created using 80-ms bins. PETHs were normalized using a z-score. For each cell its mean firing rate and SD were calculated using 20 s before and after port entry. All data are presented as normalized firing rate except for PETH of single neurons reported in *Figure 3A*. Only for display purposes, PETHs were smoothed using a 5-points moving window. A neuron was classified as responsive to a given task event if its average firing rate during the time window of interest (−640 to −80 ms before port entry, −240 to +320 ms around port exit, −200 to +200 ms around a lick) was above 1 SD of its mean firing rate. A 2 SD threshold and a time window between +160 to +400 ms relative to port entry was used to define visual cue responding cells. A neuron was considered to be active during the execution of the lick sequence (inside port cells, *Figure 3G*) if its average firing rate between 400 and 1520 ms after port entry was above 1 SD of its mean firing rate during the 40-s time window centered at port entry. Neurons whose activity during the whole waiting time period, irrespective of its duration, was above 1 SD of their mean firing rate during the 40-s time window, were considered tonically active during the waiting period (*Figure 3—figure supplement 1*). To be classified as outcome responding a neuron should met the following criteria: (1) their average firing rate between +160 to +560 ms after the eighth lick should be above 0.5 SD of their average firing rate between −640 to −160 ms prior to the eighth lick; (2) their average firing rate between +160 and +560 ms after the eighth lick should be above 2 SD of their mean activity during the 40-s time window. We further divided these neurons into three groups: (1) reward-responsive neurons (*Figure 5C*), (2) no-reward-responsive neurons (*Figure 5D*); (3) expected no-reward-responsive neurons (*Figure 5E*), if their activity in either no rewarded condition differed from activity in rewarded trials by >1.5 SD between +160 and +560 ms after the eighth lick.

To analyze the activity of port entry-responsive neurons at different waiting times (e.g., *Figure 4D–G*), PETHs were constructed using a similar number of trials for each waiting time interval, which required using waiting intervals of different duration because trial initiations were more frequent at waiting times close to the criterion wait time. Trials with waiting time shorter than 450 ms were not included in the analysis. The intervals used were (beginning, end; in ms): [451,1915; 1916,2499; 2500,3131; 3132,3803; 3804,4999; 5000,6084; 6085,8421; 8422,12621; 12622,22013; 22014,1710755]. For the data from the 2.5-s waiting time experiments presented in *Figure 7M*, the PETHs were also constructed with a different set of time intervals to increase the number of data points for shorter waiting time [451:1699, 1700:2199, 2200:2699, 2700:3199, 3200:3799, 3800:4999, 5000:6084, 6085:8421, 8422:12621, 12622:22013, 22014:1710755].

### Behavioral analysis

All data were analyzed with MATLAB software using custom-built scripts. Histograms depicting trial initiation timing at different waiting times (e.g., *Figure 2G–I*) were calculated using 100-ms bins and normalized so that the peak of the curve corresponded to a value of 1. Curves were smoothed with

**Table 1.** Detailed experimental data.

| Rat ID | Age: group/training start | Training protocol (waiting time) | Experiments | | | Recording sessions | Neurons |
|---|---|---|---|---|---|---|---|
| 7 | Adult (p118) | 2.5 s | Behavior | Electrophysiology | | 15 | 32 |
| 11 | Adult (p85) | 2.5 s | Behavior | Electrophysiology | | 7 | 62 |
| 12 | Adult (p82) | 2.5 s | Behavior | Electrophysiology | | 16 | 69 |
| 19 | Adult (p82) | 2.5 s | Behavior | Electrophysiology | | 25 | 106 |
| 29 | Adult (p128) | 2.5 s | Behavior | Electrophysiology | | 16 | 108 |
| 21 | Adolescent (p36) | 2.5 s | Behavior | Electrophysiology | | 19 | 79 |
| 25 | Adolescent (p35) | 2.5 s | Behavior | Electrophysiology | | 18 | 166 |
| 26 | Adolescent (p37) | 2.5 s | Behavior | Electrophysiology | | 13 | 307 |
| 45 | Adolescent (p42) | 2.5 s | Behavior | Electrophysiology | | 10 | – |
| 47 | Adolescent (p46) | 2.5 s | Behavior | Electrophysiology | Accelerometer | 14 | 95 |
| 49 | Adolescent (p39) | 2.5 s | Behavior | Electrophysiology | Accelerometer | 15 | 229 |
| 31 | Adult (p96) | 5 s | Behavior | Electrophysiology | | 6 | 41 |
| | | 5 s | Behavior | Electrophysiology | | 6 | 51 |
| 33 | Adult (p60) | 2.5 s | Behavior | Electrophysiology | | 2 | 25 |
| 34 | Adult (p99) | 5 s | Behavior | Electrophysiology | | 7 | – |
| 35 | Adult (p70) | 2.5–5 s | Behavior | Electrophysiology | | 22 | 279 |
| 37 | Adult (p80) | 2.5–5 s | Behavior | Electrophysiology | | 10 | – |
| 41 | Adult (p91) | 2.5–5 s | Behavior | Electrophysiology | | 9 | 94 |
| 42 | Adult (p89) | 2.5 s | Behavior | | Accelerometer | 5 | – |
| 43 | Adult (p89) | 2.5 s | Behavior | | Accelerometer | 6 | – |
| 16 | Adult (p83) | 2.5 s | Behavior | | | | |
| 17 | Adult (p83) | 2.5 s | Behavior | | | | |
| 601 | Adult (p60) | 2.5 s | Behavior | | | | |
| 602 | Adult (p60) | 2.5 s | Behavior | | | | |
| 603 | Adult (p70) | 2.5 s | Behavior | | | | |
| 604 | Adult (p70) | 2.5 s | Behavior | | | | |
| 22 | Adolescent (p35) | 2.5 s | Behavior | | | | |
| 24 | Adolescent (p33) | 2.5 s | Behavior | | | | |
| 352 | Adolescent (p36) | 2.5 s | Behavior | | | | |
| 353 | Adolescent (p28) | 2.5 s | Behavior | | | | |
| 354 | Adolescent (p30) | 2.5 s | Behavior | | | | |
| 355 | Adolescent (p30) | 2.5 s | Behavior | | | | |

a 4-bin span. Licks PETHs (e.g., *Figure 5H*) were calculated centered on port entry using a 10-ms bin and normalized to the mean and standard deviation of licks per bin for each session.

## Statistical analysis
Statistical analysis of behavioral data was performed with Prism 8, GraphPad Software. All figures were made colorblind safe using palettes from ColorBrewer 2.0 (http://colorbrewer2.org). All early

versus late comparisons were done with repeated measures two-way ANOVA, except for the reward rate in which Wilcoxon matched pairs test was used. To compare neuronal activity between Rewarded, Unrewarded, and Premature trials, repeated measures one-way ANOVA was used. To compare licking activity between Rewarded, Unrewarded, and Premature trials a mixed-effects model (REML test) was used. The performance of adults trained with a 5-s criterion time was analyzed at the late stage of training using Mann–Whitney test. Linear regression was used to correlate premature trials and the reward rate. Behavioral and neuronal data were fitted with a logistic function ($y \sim p1/(1 + \exp(-p2 * (x - p3)))$). The coefficient p1 is related to the bottom part of the curve, the coefficient p3 is related to the top part of the curve, and the coefficient p2 is related to the slope, where higher values describe steeper curves.

### Lead contact

Further information and requests for resources and reagents should be directed to and will be fulfilled by the Lead Contact, Mariano Belluscio. Behavioral and electrophysiological data and code used for analysis are reported in https://doi.org/10.5061/dryad.8kprr4xpv.

## Acknowledgements

We thank Dr. Juan Belforte for helpful discussions, Yamila Páez for her technical assistance with animal welfare, Graciela Ortega, Bárbara Giugovaz, and Germán La Iacona for technical assistance with the experiments (tetrode making and microscope image acquisition for the revised version of the manuscript). This project was funded with research grants from FONCYT (PICT 2016-0396; PICT 2017-0520; PICT 2017-2465), and UBACYT (2018-2020 305BA).

## Additional information

### Funding

| Funder | Grant reference number | Author |
|---|---|---|
| Agencia Nacional de Promoción Científica y Tecnológica | PICT 2016-0396 | Mariano Andrés Belluscio |
| Agencia Nacional de Promoción Científica y Tecnológica | PICT 2017-0520 | Mario Gustavo Murer |
| Agencia Nacional de Promoción Científica y Tecnológica | PICT 2017-2465 | Maria Cecilia Martinez |
| Secretaria de Ciencia y Tecnica, Universidad de Buenos Aires | UBACYT (2018-2020 305BA) | Mario Gustavo Murer |

The funders had no role in study design, data collection, and interpretation, or the decision to submit the work for publication.

### Author contributions

Maria Cecilia Martinez, Data curation, Software, Formal analysis, Funding acquisition, Investigation, Methodology, Writing – original draft, Writing – review and editing; Camila Lidia Zold, Methodology, Writing – original draft, Writing – review and editing; Marcos Antonio Coletti, Software, Formal analysis; Mario Gustavo Murer, Conceptualization, Resources, Formal analysis, Supervision, Funding acquisition, Writing – original draft, Project administration, Writing – review and editing; Mariano Andrés Belluscio, Conceptualization, Resources, Data curation, Software, Formal analysis, Supervision, Funding acquisition, Validation, Investigation, Visualization, Methodology, Writing – original draft, Project administration, Writing – review and editing

### Author ORCIDs

Maria Cecilia Martinez ⓘ http://orcid.org/0000-0002-1110-111X

Mario Gustavo Murer (iD) http://orcid.org/0000-0001-5149-1311
Mariano Andrés Belluscio (iD) http://orcid.org/0000-0001-5422-9992

### Ethics

All procedures complied with the National Institutes of Health Guide for Care and Use of Labora-tory Animals (Publications No. 80-23, revised 1996) and were approved by the Animal Care and Use Committee of the School of Medicine of the University of Buenos Aires – CICUAL – Resolución CD 3101/2017, RES CS-2020-1053.

### Decision letter and Author response

Decision letter https://doi.org/10.7554/eLife.74929.sa1
Author response https://doi.org/10.7554/eLife.74929.sa2

## Additional files

### Supplementary files

• Transparent reporting form

### Data availability

Behavioral and electrophysiological data and code used for analysis are reported in https://doi.org/10.5061/dryad.8kprr4xpv.

The following dataset was generated:

| Author(s) | Year | Dataset title | Dataset URL | Database and Identifier |
|---|---|---|---|---|
| Martinez MC, Belluscio M | 2022 | Dorsal striatum coding for the timely execution of action sequences | https://doi.org/10.5061/dryad.8kprr4xpv | Dryad Digital Repository, 10.5061/dryad.8kprr4xpv |

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
