## [Editor Report]

This article investigates an important topic related to the initiation signals of actions sequences detected in the dorsal striatum. The data presented convincingly support the idea that these signals distinguish between the premature versus the timely release of actions. The experiments are well-organized and substantially advance the field.

---

## [Decision Letter]

**Decision letter after peer review:**

Thank you for submitting your article "Dorsal striatum coding for the timely execution of action sequences" for consideration by *eLife*. Your article has been reviewed by 3 peer reviewers, and the evaluation has been overseen by a Reviewing Editor and Kate Wassum as the Senior Editor. The following individual involved in review of your submission has agreed to reveal their identity: Naoshige Uchida (Reviewer #1).

Essential revisions:

In general, there were concerns that while a technically impressive effort was undertaken, the data in adult subjects did not offer a significantly novel advance to the field. The adolescent data offers such potential, but was too premature with reviewers expressing concerns about technical issues and sample size. Finally, the overall clarity of the manuscript and methods needs to be attended to. Please address these and the essential revision below, as well as each point from the public reviews.

1) The difference between adult and adolescent rats are not particularly big, with the data from the adolescent rats showing a noisy trace.

2) The firing rate plot shown in Figure 4D should be replotted by aligning trials by movement initiation (presumably available from accelometer or video recording). Is it possible that the activity rise similarly between trials types but the activity is cut off depending on when the animal enters the port at different latency from the movement initiation? In any case, the port entry is a little indirect measure of "trial initiation".

3) Regarding the recording locations: if more than the DMS is recorded, the authors should revise the manuscript to reflect that and revise the citations with respect to interpreting the results as DMS-related.

4) Regarding the Ns of subjects: larger numbers of animals are needed in order to compare behavior across groups and ages.

5) The authors should use clearer and shorter sentences; they should define each phrase more clearly (i.e., be redundant in explaining what phrases like "timely" mean). There could be a table showing exactly which groups exist, exactly which task epochs are analyzed, and how each of the shorthand terms used throughout the manuscript relate to these groups and the analysis epochs.

6) The authors should make it clearer for each Results section exactly which of the groups (listed in the public review) are included in analysis.

7) I understand rats learn to execute sequences of <8licks or 8 licks, although diagrams are presented, no examples of the individual trials with 8 licks, neither distributions of bouts of these licks are presented.

8) Relevant to the statement: "in this task, the firing rate modulation preceding trial initiation discriminates between premature and timely trials and does not predict the speed, regularity, structure, value or vigor of the subsequently released action sequence"… It is not clear if the latency to first lick (plot 2D) and the inter-lick interval (2E) is only from the 8Lick sequences or not. If that is not the case, it is important to compare only the ones with 8Licks.

9) Related to the implications of the previous statement, there seems to be a tendency for longer latency to first lick in timely vs premature trials in Figure 2D (timely-trials-Late vs premature-trials-late)? Again here it is important to compare the 8licks sequences only.

10) I could not find in the main text whether the individual points in Figure 2 (e.g. 2B-E) are individual animals. Please specify that.

11) Although very elegant the argument presented in Figure 4C and 6C, I wonder if the head acceleration may lose differences in movements outside the head in the two kinds of trials. If that is the case please acknowledge it.

12) Also in 4C, small separations between timely vs premature signals are seen before 0. Is there a way to know if animals in timely vs premature trials approached the entry port in the same way? This request is pertinent in order to rule out motor contribution to the differences in Figure 4A-B.

13) When saying: "Similar results were obtained in rats trained with a longer waiting interval (Supplementary Figure 5)", "is hard to see the similarity in the premature range, while in the 2.5 seconds task there is a positive relationship in the 5 seconds task it is not.

14) The data showing that the waiting modulation of reward anticipation grows at a faster rate in adolescent rats is clear, however, it is not clear how it could be related to the data showing that the adolescent rats were more impulsive.

15) Related to the sentence: "the strength of anticipatory activity increased with the time waited before response release and was higher in the more impulsive adolescent rats"….One may expect to see a difference in the range of the premature time however the differences were observed in the range >2.5 seconds. Please explain how to reconcile this finding with the fact that the adolescent rats were more impulsive.

16) Please ensure your manuscript complies with the *eLife* policies for statistical reporting: https://reviewer.elifesciences.org/author-guide/full "Report exact p-values wherever possible alongside the summary statistics and 95% confidence intervals. These should be reported for all key questions and not only when the p-value is less than 0.05."

17) Please state that male rats were the subjects in the abstract and in your discussion discuss the limitations of the exclusions of females from this study.

---

## [Author Response]

Essential revisions:In general, there were concerns that while a technically impressive effort was undertaken, the data in adult subjects did not offer a significantly novel advance to the field. The adolescent data offers such potential, but was too premature with reviewers expressing concerns about technical issues and sample size. Finally, the overall clarity of the manuscript and methods needs to be attended to. Please address these and the essential revision below, as well as each point from the public reviews.

While the significance of the striatal activity that precedes or accompanies the initiation of a learned action sequence is not completely understood, in particular regarding the timing of action sequence initiations, we acknowledge that the data collected from adolescent animals are particularly interesting and have included new data and analysis to further understand the observed differences between adolescents and adults.

(1) The difference between adult and adolescent rats are not particularly big, with the data from the adolescent rats showing a noisy trace.

New data from two adolescent rats reduced the variability and confirmed the behavioral and physiological differences with adult rats. All panels from Figure 7 and Figure 7- supplements 1-3 now include the data from 5 adolescent animals instead of 3. The number of neurons analyzed in the adolescent group passed from 552 to 876. The inclusion of these new data allowed us to perform new statistical comparisons. We adjusted a logistic function to accumulated trial initiation timing data (Figure 7N) and found that the rate of accumulation is higher in adolescent rats. Importantly, this is observed not only in the part of the curve corresponding to premature responding but also during timely responding, indicating that adolescent rats' premature responding is a manifestation of a more general behavioral trait that makes them self-initiate trials faster than adults (Figure 7N). The noisy trace of curves showing the amplitude modulation of anticipatory activity as a function of waiting time was partly due to the relatively low number of premature trials that demanded using relatively long time bins. With more data available we have been able to replot these curves using a smaller bin size for the short waiting times (Figure 7M). We have adjusted a logistic function to these data and observed a higher rate of increase of this activity modulation in adolescent rats, paralleling the behavioral data. Moreover, we report a significant correlation between the behavioral and neurophysiological data (a steeper rate of trial initiation times curve correlates with a steeper wait modulation of anticipatory activity, Figure 7O).

(2) The firing rate plot shown in Figure 4D should be replotted by aligning trials by movement initiation (presumably available from accelometer or video recording). Is it possible that the activity rise similarly between trials types but the activity is cut off depending on when the animal enters the port at different latency from the movement initiation? In any case, the port entry is a little indirect measure of "trial initiation".

Unfortunately, we have not systematically obtained video recordings of the sessions and only have accelerometer recordings of a few of the animals that provided the neuronal data, which precludes replotting the data as suggested. Accelerometer recordings are available from two adult and two adolescent rats. Latency from movement initiation to port entry does not differ between premature and timely trials at both ages. This is now reported in the results section. These results appear to be at odds with the idea that decreased neuronal activity in premature trials is the result of a cut-off of the response.

3) Regarding the recording locations: if more than the DMS is recorded, the authors should revise the manuscript to reflect that and revise the citations with respect to interpreting the results as DMS-related.

We acknowledge that our recordings are spread along the medial and central regions of the dorsal striatum. Although we are not sure that there is a consensus regarding the limits of the DMS and DLS, we believe that none of our recordings are clearly located within the DLS. Following your suggestion, we have modified the text and refer to the location of our recordings as “dorsal striatum”. We believe that, as there is a lot of work on the roles of the DLS and DMS in reward learning, it is still important to refer to this work in the Introduction section and to discuss our findings in its context, particularly, since we find that most task-related activity is concentrated at the beginning and end of the task as shown in several studies focused in the DLS. Nevertheless, we removed the supplementary figures comparing our more medial and lateral recordings to avoid a misinterpretation of recording locations.

(4) Regarding the Ns of subjects: larger numbers of animals are needed in order to compare behavior across groups and ages.

Two adolescent animals provided new behavioral and electrophysiological data, and one additional adolescent rat with implanted electrodes provided behavioral data but no useful neuronal recordings. New behavioral data from non-implanted 6 adult and 6 adolescent rats confirming the results obtained with the implanted animals are presented in Figure 7 – supplement 3. The number of animals used in each experiment and contributing to each figure is now reported in Table 1.

(5) The authors should use clearer and shorter sentences; they should define each phrase more clearly (i.e., be redundant in explaining what phrases like "timely" mean). There could be a table showing exactly which groups exist, exactly which task epochs are analyzed, and how each of the shorthand terms used throughout the manuscript relate to these groups and the analysis epochs.

The manuscript was thoroughly revised and Table 1 summarizes the requested information. In addition, we included information in each figure legend to clarify from which animal groups and tasks the data come.

(6) The authors should make it clearer for each Results section exactly which of the groups (listed in the public review) are included in analysis.

As mentioned in the previous point, we now include Table 1 summarizing which group is used in each analysis.

(7) I understand rats learn to execute sequences of <8licks or 8 licks, although diagrams are presented, no examples of the individual trials with 8 licks, neither distributions of bouts of these licks are presented.

Rats learn to execute a lick sequence to obtain the reward. The experiments do not allow us to establish if they know what the exact number of licks needed is; when the skill is acquired, licking becomes highly stereotyped and the rats might as well be learning a time after which continuous licking leads to reward. A representative raster plot showing lick sequences in a session in a trained adult rat is presented in Figure 1I and Figure 7 – Supplement 1H shows an example of the licks of an adolescent rat.

(8) Relevant to the statement: "in this task, the firing rate modulation preceding trial initiation discriminates between premature and timely trials and does not predict the speed, regularity, structure, value or vigor of the subsequently released action sequence"… It is not clear if the latency to first lick (plot 2D) and the inter-lick interval (2E) is only from the 8Lick sequences or not. If that is not the case, it is important to compare only the ones with 8Licks.

The data are from 8 lick sequences, this is now indicated in the figure legend.

(9) Related to the implications of the previous statement, there seems to be a tendency for longer latency to first lick in timely vs premature trials in Figure 2D (timely-trials-Late vs premature-trials-late)? Again here it is important to compare the 8licks sequences only.

Only 8-lick sequences are compared and the two-way ANOVA showed a significant effect of the training stage without significant effects of trial type and a non-significant interaction. The average ± SEM latencies to the first lick (of the eighth lick sequence) were 0.717 s ± 0.063 for timely trials late and 0.805 s ± 0.086 for premature trials late.

(10) I could not find in the main text whether the individual points in Figure 2 (e.g. 2B-E) are individual animals. Please specify that.

Every individual point corresponds to the mean of a session, the data correspond to 5 adult animals (2-5 sessions per animal and timing condition). Whether the data correspond to animals or sessions is now clarified in all figure legends.

(11) Although very elegant the argument presented in Figure 4C and 6C, I wonder if the head acceleration may lose differences in movements outside the head in the two kinds of trials. If that is the case please acknowledge it.

We acknowledge in the main text, Results section, that the accelerometer does not allow us to determine if the movements of other body parts differ between trial types.

(12) Also in 4C, small separations between timely vs premature signals are seen before 0. Is there a way to know if animals in timely vs premature trials approached the entry port in the same way? This request is pertinent in order to rule out motor contribution to the differences in Figure 4A-B.

Although it is not possible to completely rule out small movement differences between premature and timely trials, no evident behavioral differences can be detected by trained observers or by analyzing video recordings taken during some sessions. The available accelerometer recordings also suggest that a similar motor pattern is displayed in premature and timely trials (Figure 4C).

(13) When saying: "Similar results were obtained in rats trained with a longer waiting interval (Supplementary Figure 5)", "is hard to see the similarity in the premature range, while in the 2.5 seconds task there is a positive relationship in the 5 seconds task it is not.

Please note that a positive relationship is observed for the two bins preceding trial initiation, which are about 2.75s and 1s before port entry. The bin that seems to not fit is centered 4s before port entry (1s after exiting the port in the previous trial). Because of the longer waiting time behavior becomes less organized during the first seconds after port exit in the 5s task, however, the modulation of activity is still observed in the bins that are close to port entry.

(14) The data showing that the waiting modulation of reward anticipation grows at a faster rate in adolescent rats is clear, however, it is not clear how it could be related to the data showing that the adolescent rats were more impulsive.

We acknowledge that the data do not provide a causal link with behavior. After adding two new adolescent rats we have been able to study in more detail the relationship between the waiting modulation of neuronal activity and the accumulation of trial initiations (depicted in figures 7M and 7N respectively) by adjusting logistic functions to the data. There is a striking parallel between the growth rate of both curves, and the curves of adolescent rats are significantly steeper than those of adult rats. Moreover, there is a significant correlation between the coefficients that mark the rate of growth of the behavioral and neurophysiological data (Figure 7O).

(15) Related to the sentence: "the strength of anticipatory activity increased with the time waited before response release and was higher in the more impulsive adolescent rats"….One may expect to see a difference in the range of the premature time however the differences were observed in the range >2.5 seconds. Please explain how to reconcile this finding with the fact that the adolescent rats were more impulsive.

Please, note that the more impulsive behavior of adolescent rats (and the faster growth of the wait modulation of anticipatory activity) is observed along waiting times that exceed the 2.5s criterion wait time; we added a phrase in the Results section and in the Discussion section to emphasize this point. Regarding the premature trials, we have used smaller bins to better discriminate what happens at short waiting times and included an inset in Figure 7M that allows us to better appreciate these intervals.

(16) Please ensure your manuscript complies with the eLife policies for statistical reporting: https://reviewer.elifesciences.org/author-guide/full "Report exact p-values wherever possible alongside the summary statistics and 95% confidence intervals. These should be reported for all key questions and not only when the p-value is less than 0.05."

Exact p-values are now reported.

(17) Please state that male rats were the subjects in the abstract and in your discussion discuss the limitations of the exclusions of females from this study.

We added a sentence in the discussion, as requested, and stated that males were used in the abstract.